# Directed Exploration in Reinforcement Learning from Linear Temporal Logic

**Marco Bagatella**                                                     *mbagatella@ethz.ch*
*Department of Computer Science*
*ETH Zürich, Zürich, Switzerland*

**Andreas Krause**
*Department of Computer Science*
*ETH Zürich, Zürich, Switzerland*

**Georg Martius**
*Max Planck Institute for Intelligent Systems*
*Tübigen, Germany*

**Reviewed on OpenReview:** *https://openreview.net/forum?id=cjK5ZvP4zZ*

## Abstract

Linear temporal logic (LTL) is a powerful language for task specification in reinforcement learning, as it allows describing objectives beyond the expressivity of conventional discounted return formulations. Nonetheless, recent works have shown that LTL formulas can be translated into a variable rewarding and discounting scheme, whose optimization produces a policy maximizing a lower bound on the probability of formula satisfaction. However, the synthesized reward signal remains fundamentally sparse, making exploration challenging. We aim to overcome this limitation, which can prevent current algorithms from scaling beyond low-dimensional, short-horizon problems. We show how better exploration can be achieved by further leveraging the LTL specification and casting its corresponding Limit Deterministic Büchi Automaton (LDBA) as a Markov reward process, thus enabling a form of high-level value estimation. By taking a Bayesian perspective over LDBA dynamics and proposing a suitable prior distribution, we show that the values estimated through this procedure can be treated as a shaping potential and mapped to informative intrinsic rewards. Empirically, we demonstrate applications of our method from tabular settings to high-dimensional continuous systems, which have so far represented a significant challenge for LTL-based reinforcement learning algorithms.

## 1 Introduction

Most reinforcement learning (RL) research has traditionally focused on a standard setting, prescribing the maximization of cumulative rewards in a Markov Decision Process (Puterman, 2014; Sutton & Barto, 2018). While this simple formalism captures a variety of behaviors (Silver et al., 2021), its expressiveness remains limited (Abel et al., 2021). In pursuit of a more natural and effective way to specify desired behavior, several works have turned towards logic languages (Camacho et al., 2019; De Giacomo et al., 2020b; Hasanbeig et al., 2018; Icarte et al., 2022; Li et al., 2017). Originally designed to describe possible paths of a system (with direct applications, e.g., in model checking (Baier & Katoen, 2008)), Linear Temporal Logic (LTL) (Pnueli, 1977) has been found to strike an interesting balance between expressiveness and tractability.

A translation from LTL to reinforcement learning objectives is in general possible at the cost of optimality (Alur et al., 2022). Nevertheless, several works (Bozkurt et al., 2020; Hasanbeig et al., 2020; Voloshin et al.,

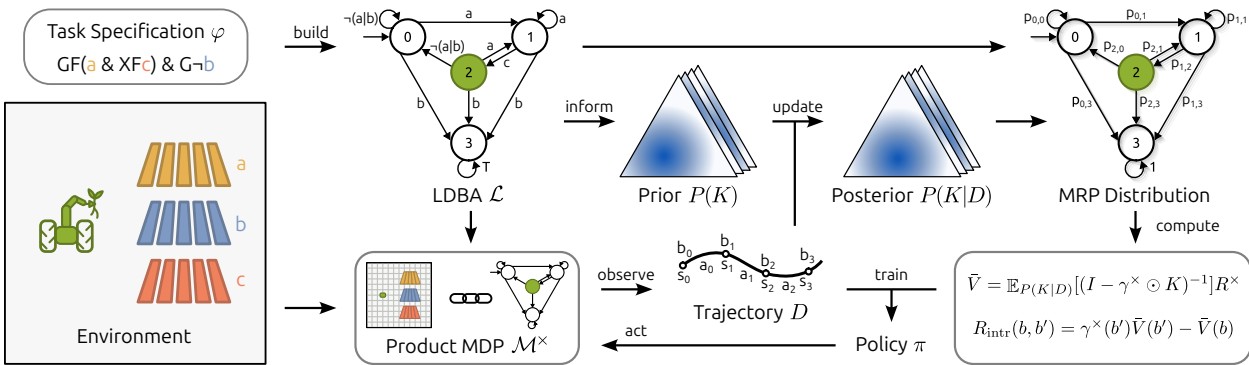

Figure 1: Overview of DRL$^2$. DRL$^2$ leverages an LDBA representation of the task and a Bayesian estimate of its transition kernel to define a distribution over Markov reward processes, which can be used for high-level value estimation. Resulting values are mapped to an intrinsic reward signal, which guides exploration in the product MDP.

2023) have proposed a reward and discounting scheme to distill a policy through reinforcement learning from an LTL specification. In particular, the scheme proposed by Voloshin et al. (2023) recovers a policy that optimizes a lower bound on the probability of satisfying the specification: suboptimality can in practice be bounded through discretization arguments, or for finite policy classes. However, this scheme results in a *sparse* reward signal and a potentially *flat* value landscape, thus making exploration a fundamental challenge.

The necessity for strong exploration algorithms when learning from LTL specification is therefore evident. Existing methods rely on counterfactual data augmentation schemes (Voloshin et al., 2023), which however do not directly guide the agent in the underlying MDP, on the availability of a task distribution (Wang et al., 2023), or on heuristics (Hasanbeig et al., 2020). Closer in spirit to our work, task-aware reward shaping has briefly been explored in the context of finite logic (Camacho et al., 2019; Icarte et al., 2022), but has not been scaled to $\omega$-regular problems and cannot adapt to unknown environment dynamics.

The main idea of our work is the distillation of an intrinsic reward signal from the structure of the LTL specification itself. In particular, we repurpose the Limit Deterministic Büchi Automata (LDBA) constructed from an LTL formula as a Markov reward process, by assuming a transition kernel over LDBA states. We then leverage the reward process to perform a form of high-level value estimation and compute values for given LDBA states. These values can be naturally leveraged for potential-based reward shaping. Crucially, we adopt a Bayesian perspective on estimating the transition kernel over the LDBA: by choosing a suitable prior distribution, we ensure that intrinsic rewards are informative from the initial phases of learning. This is done by *optimistically* assuming that the agent is capable of transitioning to any adjacent LDBA state in a single step, although this might not be easily afforded by the dynamics of the environment. Moreover, by updating the distribution according to evidence, the assumed transition kernel can be adapted over time to realistically represent the agent's behavior and environment's dynamics.

Our method, named DRL$^2$ (**D**irected **R**einforcement **L**earning from Linear Temporal **L**ogic), is illustrated in Figure 1 and is capable of driving deep exploration in reinforcement learning from LTL specifications. The contributions of this work can be outlined as follows:

1. we design and introduce a method for exploration in reinforcement learning when learning from LTL specifications, by casting the LDBA as a Markov reward process and leveraging it for value estimation and distillation of intrinsic rewards;
2. we analyse the proposed method and the suboptimality potentially induced by intrinsic rewards;
3. we evaluate the method across diverse settings, spanning from simple tabular cases to complex, high-dimensional environments, which pose a significant challenge for reinforcement learning from LTL specifications.

Section 2 provides an introduction to LTL and its connection to reinforcement learning. Our method is described in Section 3 and evaluated in Section 4. A discussion of related works and of the proposed one can be found in Sections 5 and 6, respectively.

## 2 Background

This section provides a brief introduction to LTL and discusses connections to reinforcement learning. For a complete introduction, we refer the reader to Baier & Katoen (2008).

**Linear Temporal Logic**  LTL formulas build upon a finite set of propositional variables (atomic propositions, AP), over which an alphabet is defined as the powerset $\Sigma = 2^{\text{AP}}$, i.e. the combinations of variables evaluating to `true`.

As a more concrete, illustrative example, let us consider a farming robot, which is tasked with continually weeding through any of three different fields. The presence of the robot in each field could be described by the set of atomic propositions $\{a, b, c\}$. When the robot is operating in the first field, $a$ would evaluate to `true` ($\top$), while $b$ and $c$ would evaluate to `false` ($\bot$).

**Definition 2.1.** (LTL Formula) An LTL formula is a composition of atomic propositions (AP), logical operators `not` ($\neg$) and `or` ($\,|\,$) and temporal operators `next` ($X$) and `until` ($U$). Inductively:

- if $p \in AP$, then $p$ is an LTL formula;
- if $\phi$ and $\theta$ are LTL formulas, then $\neg\phi$, $\phi\,|\,\theta$, $X\,\phi$ and $\phi\,U\,\theta$ are LTL formulas.

Intuitively, while logical operators encode their conventional meaning, the `next` operator evaluates to `true` if its argument holds true at the very next time step, and `until` is a binary operator which requires its second argument to eventually evaluate to `true`, and its first argument to hold true until this happens. From this sufficient set of operators, additional ones are often defined in order to allow more concise specifications. In the context of reinforcement learning for control, useful operators are `finally` ($F(\phi) := \top U \phi$) and `globally` ($G(\varphi) := \neg F \neg \phi$). Returning to our example, they could be used to specify stability ($FGa$, i.e., reach and remain in the first field), or avoidance ($G\neg a$, i.e., never enter the first field). A more complex task, which requires the farming robot to visit the first and third fields, repeatedly, while always avoiding the second one, could be simply represented as $(GF(a\&XFc))\&(G\neg b)$. Through this work, we will refer to this task as $T_0$.

Each LTL formula can be *satisfied* by an infinite sequence of truth evaluations of AP (i.e., an $\omega$-*word*). While a direct definition is also possible (Thomas, 1990), for simplicity, satisfaction will be introduced through the acceptance of paths in an automaton built from the formula.

**From Formulas to Automata**  A practical way of defining satisfaction involves the introduction of Limit Deterministic Büchi Automata (LDBAs). An LDBA can be constructed from any LTL formula (Sickert et al., 2016) and is able to keep track of the progression of its satisfaction.

**Definition 2.2.** (Limit Deterministic Büchi Automaton - LDBA) An LDBA is a tuple $\mathcal{L} = (\mathcal{B}, \Sigma, P^{\mathcal{B}}, \mathcal{B}^{\star}, b_0)$, where $\mathcal{B}$ is a finite set of states, $\Sigma$ is an alphabet, $P^{\mathcal{B}} : \mathcal{B} \times \Sigma \to 2^{\mathcal{B}}$ is a non-deterministic transition function, $\mathcal{B}^{\star} \subseteq \mathcal{B}$ is a set of accepting states, and $b_0 \in \mathcal{B}$ is the initial state. There exists a mutually exclusive partitioning of $\mathcal{B} = \mathcal{B}_D \cup \mathcal{B}_N$ such that $\mathcal{B}^{\star} \subseteq \mathcal{B}_D$ and for $(b, a) \in (\mathcal{B}_D \times \Sigma)$ then $P^{\mathcal{B}}(b, a) \subseteq \mathcal{B}_D$ and $|P^{\mathcal{B}}(b, a)| = 1$.

In LDBAs synthesized from LTL formulas, additional properties hold (Sickert et al., 2016). First, the alphabet is over evaluations of atomic propositions $\Sigma = 2^{AP}$. Second, the non-determinism is limited to a subset of so-called $\epsilon$-transitions, which arise for specific formulas, e.g. those encoding stability problems. As a consequence, this class of LDBAs is known as *Good-for-MDPs* (Hahn et al., 2020), since the non-determinism can be resolved on the fly in arbitrary MDPs without changing acceptance.

An infinite sequence of LDBA *actions* $(a_i)_0^{\infty} \in \Sigma^{\infty}$ induces multiple *paths*, where a single path can be defined as a sequence of LDBA states $p = (b_i)_0^{\infty}$. The set of induced paths $P_{\infty}$ can be defined recursively: $P_0 = \{(b_0)\}$ and $P_i = \{(b_0, \ldots, b_{i-1}, b_i) | (b_0, \ldots, b_{i-1}) \in P_{i-1}, b_i \in P^{\mathcal{B}}(b_{i-1}, a_i)\}$. Intuitively, each path may split if the selected LDBA action transitions to multiple states.

**Definition 2.3.** (Acceptance) An LDBA $\mathcal{L}$ accepts a path $(b_i)_0^{\infty}$ if and only if the path visits an accepting state $b^{\star} \in \mathcal{B}^{\star}$ infinitely often, that is $\forall t \in \mathbb{N}, \exists t' > t$ such that $b_{t'} \in \mathcal{B}^{\star}$.

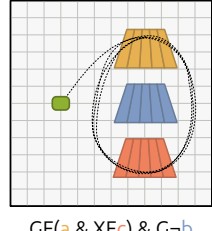 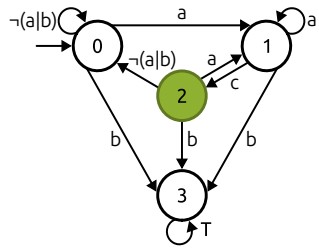

GF(a & XFc) & G¬b

Figure 2: Illustrative task $T_0$ (left), and LDBA encoding the formula (right, with starting state marked as 0 and accepting state in green). The farming robot (in green) moves in a 2D plane, where three areas of different colors represent fields. $\{a, b, c\}$ are atomic propositions that evaluate to `true` when the agent enters the yellow, blue and red field, respectively. A trajectory satisfying the specification is shown.

By translating each LTL specification into an LDBA, it is now possible to tie formula satisfaction to the acceptance of a path: informally, an infinite sequence of AP evaluations ($\omega$-*word*) satisfies a formula $\varphi$ if and only if the LDBA $\mathcal{L}$ synthesized from $\varphi$ accepts a path $(b_i)_0^\infty$ induced by the $\omega$-word.

Let us consider the example in Figure 2 for the illustrative task $T_0$ and the LTL formula $\varphi = (GF(a\&XFc))\&(G\neg b)$. The periodic LDBA path $(0, 1, 2)^\infty$ is accepted, just as $(0, 1, 1, 2)^\infty$, although the latter takes on average more steps to reach the accepting state. On the other hand, the paths $(0)^\infty$ or $(0, (3)^\infty)$ would not be accepted, as neither ever reach the accepting state, with the second getting caught in a sink state by violating the avoidance criterion in $\varphi$.

We note that this notion of success relies on conditions that are achieved *eventually* in the future, independently of temporal distance. While this leads to myopic behavior under naive optimization (Voloshin et al., 2023), recent works (Bozkurt et al., 2020; De Giacomo et al., 2020a; Hasanbeig et al., 2020; Voloshin et al., 2023) propose an elegant rephrasing for formula satisfaction in an RL-friendly form, as we now describe.

**Product MDPs and Policy Optimization**  The following part describes standard reinforcement learning terminology and then reconciles it with the introduced logic machinery.

**Definition 2.4.** (Markov Decision Process – MDP) A Markov Decision Process is a tuple $\mathcal{M} = (\mathcal{S}, \mathcal{A}, P, R, \gamma, \mu_0)$, where $\mathcal{S}$ and $\mathcal{A}$ are potentially continuous state and action spaces, $P : \mathcal{S} \times \mathcal{A} \to \Delta(\mathcal{S})$ is a probabilistic transition kernel[1], $R : \mathcal{S} \times \mathcal{A} \to \mathbb{R}$ is a reward function, $\gamma$ is a discount factor, and $\mu_0 \in \Delta(\mathcal{S})$ is the initial state distribution.

MDPs are the standard modeling choice for reinforcement learning environments; however, they are disconnected from atomic propositions and unable to track progression over formula satisfaction. The semantics of APs can be grounded in MDP states through a labeling function $\mathcal{F} : \mathcal{S} \to \Sigma$, which evaluates APs in each MDP state. On the other hand, progression over task satisfaction can naturally be stored in an LDBA, as its states encode sufficient information on the history of paths.

Finally, all three components (MDP, labeling function and LDBA) can be *synchronized* to ensure consistency between trajectories in each of them and enable mapping policies to distribution over paths, and therefore likelihoods of formula satisfaction (Hasanbeig et al., 2018; Voloshin et al., 2023). The first step is to resolve the non-determinism by creating a new action for each potential $\epsilon$-transition to a state $b \in \mathcal{B}$, thus defining a new action set $\mathcal{A}^{\mathcal{B}} = \{\epsilon_b | b \in \mathcal{B}\}$ (Sickert et al., 2016). Then, we can define a product MDP, encompassing the underlying MDP and the LDBA.

**Definition 2.5.** (Product MDP) Given an MDP $\mathcal{M}$ and a LDBA $\mathcal{L}$, their Product MDP is a tuple $\mathcal{M}^\times = (\mathcal{S}^\times, \mathcal{A}^\times, \mathcal{P}^\times, R^\times, \gamma^\times, \mu_0^\times)$, where $\mathcal{S}^\times = \mathcal{S} \times \mathcal{B}, \mathcal{A}^\times = \mathcal{A} \cup \mathcal{A}^{\mathcal{B}}$, and $\mu_0^\times(s, b) = \mu_0(s) \cdot \mathbf{1}_{b=b_0}$. Furthermore,

$$P^\times((s', b')|a, (s, b)) = \begin{cases} P(s'|a, s), & a \in \mathcal{A}, b' \in P^{\mathcal{B}}(b, \mathcal{F}(s')) \\ 1 & a \in \mathcal{A}^{\mathcal{B}}, a = \epsilon_{b'}, b \xrightarrow{\epsilon} b', s = s' \\ 0 & \text{otherwise,} \end{cases}$$

---

[1] $\Delta(\mathcal{S})$ represents the space of probability distributions over $\mathcal{S}$.

where $b \xrightarrow{\epsilon} b'$ indicates that there is an $\epsilon$-transition between the two LDBA states $b$ and $b'$. Finally, the reward and discount factor are defined as

$$R^\times(b_i) = \begin{cases} 1, & \{b_i \in \mathcal{B}^*\} \\ 0, & \text{otherwise} \end{cases}, \quad \gamma^\times(b_t) = \begin{cases} \gamma, & b_t \in \mathcal{B}^* \\ 1, & \text{otherwise.} \end{cases} \tag{1}$$

Essentially, a policy learned over the product MDP has access to both low-level information (MDP states) and indicators of progress on the specified task (LDBA states). The transition kernel over $\mathcal{M}^\times$ guarantees that both the underlying MDP and the LDBA evolve consistently. This synchronization is achieved through the labeling function $\mathcal{F}$. Through this construction, it is finally possible to connect trajectories (and, therefore, policies) to satisfaction of a given LTL formula. Let us consider an LTL formula $\varphi$ and its corresponding LDBA $\mathcal{L}$, as well as a trajectory $\tau = (s_i, b_i)_0^\infty$ in the product MDP $\mathcal{M}^\times$. Then, $\tau \models \varphi$ ($\tau$ satisfies $\varphi$) if and only if $\mathcal{L}$ accepts the path $(b_i)_0^\infty$, i.e., the projection of $\tau$ to LDBA states. Finally, let us consider a policy $\pi : \mathcal{S}^\times \to \Delta(\mathcal{A}^\times)$: the probability of $\pi$ satisfying $\varphi$ can thus be defined as the probability integral for trajectories satisfying the formula: $P(\pi \models \varphi) = \mathbb{E}_{\tau \sim \pi} \mathbf{1}_{\tau \models \varphi}$. Optimizing a policy $\pi$ for satisfaction of an LTL specification $\varphi$ can be expressed as finding $\pi^\star \in \text{argmax}_{\pi \in \Pi} P(\pi \models \varphi)$. Prior works (Hasanbeig et al., 2018; Voloshin et al., 2023) have proposed RL-friendly proxy objectives, which optimize a lower bound on the probability of LTL formula satisfaction:

$$\pi_\gamma^\star \in \underset{\pi \in \Pi}{\text{argmax}} \; \underset{\tau \sim \pi}{\mathbb{E}} \left[ \sum_{i=0}^\infty \Gamma_i R^\times(b_i) \right] \; (:= V_\pi^\gamma), \quad \text{where } \Gamma_0 = 1, \; \Gamma_i = \prod_{t=0}^{i-1} \gamma^\times(b_t). \tag{2}$$

Paraphrasing, $\pi_\gamma^\star$ maximizes the visitation count to accepting states in the LDBA under *eventual discounting*. For a formal analysis of the policy recovered by this objective, we refer the reader to Voloshin et al. (2023). Our work builds upon this formulation and devises an exploration method to compensate for its drawbacks. That is, the reward function $R^\times$ is fundamentally sparse: the agent only receives feedback when a significant amount of progress toward solving the task has been made, and thus an accepting LDBA state is visited. As a result, while naive exploration might reach several non-accepting LDBA states, the agent remains unable to evaluate them, as it is largely uninformed of the yet unexplored parts of the LDBA. Our method distills the global known structure of the LDBA in a denser intrinsic reward signal for exploration.

## 3 Method: Directed Exploration in Reinforcement Learning from LTL

Our method relies on (i) repurposing the LDBA as a Markov reward process by assigning a transition kernel, as well as rewards and discount signals for each transition, (ii) defining a value estimation operator to compute high-level values for each LDBA state, and finally (iii) leveraging these values for potential-based reward shaping. This procedure is described in Section 3.1. It takes an LDBA and a transition kernel as input and returns intrinsic rewards for each transition in the product MDP. A second and crucial component, discussed in Section 3.2, is the estimation of the LDBA transition kernel, which is essential for ensuring informative intrinsic rewards: by taking a Bayesian perspective, we show that a symmetric prior can induce strong exploration. Finally, Section 3.3 connects each component and describes a practical instantiation of the algorithm.

### 3.1 High-level Value Estimation from LDBA

As described in Section 2, an LDBA can be naturally synthesized from a given LTL specification, through well-known schemes (Sickert et al., 2016). We now show how an LDBA can be recast as a Markov reward process, which can be leveraged for computing values for each LDBA state. This construction requires 2 ingredients, namely (i) an LDBA $\mathcal{L}$, and (ii) a transition kernel $K$ over LDBA states, where $K$ is a stochastic matrix such that $K_{i,j} = \mathbb{E}_{\pi, P^\times, \mu_0^\times} P(b' = b_j | b = b_i)$, i.e., an estimate of the probability for the LDBA to transition to state $b_j$ starting from $b_i$ under some policy $\pi$, assuming stationarity. The choice of the transition kernel $K$ is crucial for the effectiveness of the method and is thus treated in detail in the following section.

Having access to these two ingredients, we can define a discrete Markov reward process $\bar{\mathcal{L}} = (\mathcal{B}, K, R^\times, \gamma^\times, b_0)$: the state space $\mathcal{B}$ and initial state $b_0$ are left unchanged and coupled with the transition kernel $K$. Moreover,

we provide reward and a discounting functions: $R^\times$ and $\gamma^\times$ are a projection of the eventual reward and discounting scheme to the LDBA, as they are only dependent on LDBA states (see Equation 1). A trajectory $\tau = (b_0, b_1, \dots)$ can be sampled from the MRP according to $p(\tau) = \prod_{i=1}^\infty K_{b_{i-1}, b_i}$. As any Markov reward process, the newly synthesized one allows computing the value function under eventual discounting $\bar{V}_K(b) = \mathbb{E}_K[\sum_{t=0}^\infty \Gamma_t R^\times(b_t)]$ with $b_{t+1} \sim K(b_t)$. The Markov property over the MRP induces the following Bellman equation (Sutton & Barto, 2018):

$$\bar{V}_K = R^\times + \gamma^\times \odot (K\bar{V}_K), \tag{3}$$

where a matrix notation is adopted: $\bar{V}_K, R^\times$ and $\gamma^\times$ are represented as $|\mathcal{B}|$-dimensional vectors, and $\odot$ stands for the Hadamard product. As the LDBA (and thus the MRP) is discrete and finite [2], the Bellman equation defined over the MRP has a closed-form solution (Sutton & Barto, 2018) [3]:

$$\bar{V}_K = (\mathbb{I} - \gamma^\times \odot K)^{-1} R^\times, \tag{4}$$

where $\mathbb{I}$ represents the identity matrix. As the MRP is closely related to the product MDP $\mathcal{M}^\times$, an analysis of their connection is provided in Appendix B. Once value estimates $\bar{V}_K$ are computed, they can be treated as a potential function for reward shaping (Ng et al., 1999), although under eventual discounting (Voloshin et al., 2023):

$$R_{\text{intr}}(b, b') = \gamma^\times(b')\bar{V}_K(b') - \bar{V}_K(b). \tag{5}$$

This reward signal can be added to the product MDP reward $R^\times$ and optimized with an arbitrary reinforcement learning algorithm. We note that, as an instantiation of potential-based reward shaping, the optimal policy in the product MDP remains invariant to this reward transformation, if eventual discounting converges to zero.

**Proposition 3.1.** *(Consistency) Let us consider the product MDP $\mathcal{M}^\times = (\mathcal{S}^\times, \mathcal{A}^\times, \mathcal{P}^\times, R^\times, \gamma^\times, \mu_0^\times)$ and its modification $\widetilde{\mathcal{M}}^\times = (\mathcal{S}^\times, \mathcal{A}^\times, \mathcal{P}^\times, R^\times + R_{intr}, \gamma^\times, \mu_0^\times)$. Under eventual discounting, any optimal policy in $\widetilde{\mathcal{M}}^\times$ for which $\Gamma_t \overset{t \to \infty}{\to} 0$ is also optimal in $\mathcal{M}^\times$.*

The proof follows the general scheme for potential-based reward shaping (Ng et al., 1999) and extends it to the eventual discounting setting (see Appendix A).

While this procedure allows the synthesis of an intrinsic reward signal for arbitrary logic specifications, the informativeness of this signal relies on two factors. The first factor is the existence of a sink state, which occurs across many (but not all) LDBAs constructed from LTL formulas (e.g., formulas involving global avoidance, as the illustrative task $T_0$). Its absence can be remedied by augmenting the MRP state space with a virtual sink state, reachable from all other states and associated with an eventual reward and discount factors of 0 and 1, respectively (see Appendix C for a complete discussion and evaluation). The second factor lies in the transition kernel $K$. While Proposition 3.1 guarantees that no kernel perturbs the optimal policy, it does not quantify how a chosen kernel affects learning efficiency. The following section outlines how a careful choice of its initialization and estimation is crucial to the practical effectiveness of the algorithm. Other alternatives are ablated empirically in Appendix F.

## 3.2 Optimistic Priors for High Level Value Estimation

Let us consider a naive approach to the choice of the transition kernel $K$, which simply computes the expected empirical transition probability of the current policy $\pi$. As shown at the top of Figure 3 for the illustrative task $T_0$, a randomly initialized policy can be executed in the product MDP, and $K$ can be estimated through the empirical count of observed transitions in the LDBA. As long as the policy does not spontaneously visit the accepting state, values and rewards estimated through Equations 4 and 5 are uniformly zero and fail to drive exploration. The method would thus be rendered ineffective.

In order to address this issue, we adopt a Bayesian approach to the problem of estimating the LDBA transition kernel $K$. Let us consider each row $K_i$, which models a categorical probability distribution over future

---

[2]We remark that the MRP is constructed from the LDBA, which is discrete in nature, and not from the underlying MDP, which can be potentially continuous.

[3]In the case of exceedingly complex specifications, and thus large LDBAs, iterative methods could be a viable replacement.

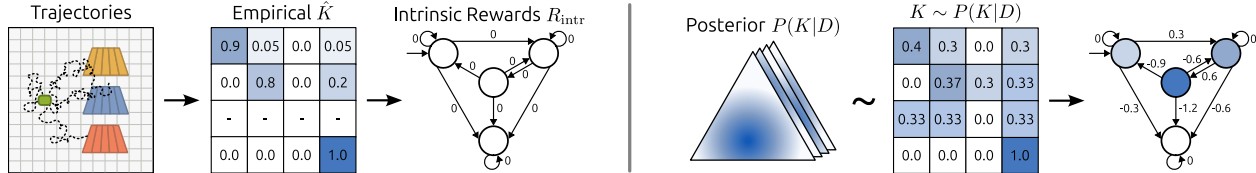

Figure 3: Left: a randomly initialized policy is executed in the product MDP $\mathcal{M}^\times$ for T0; its empirical transition kernel $\hat{K}$ results in uniformly zero value estimates $\bar{V}_K$. Right: the expected value for $K$ over a posterior distribution estimated from a symmetric prior results in informative value estimates and intrinsic rewards.

LDBA states from each LDBA state $b_i \in \mathcal{B}$. At its core, our method proposes a prior distribution over these categoricals, such that appropriate shaping of the prior controls and directs the exploration in the product MDP. As the conjugate prior to categorical distributions, we adopt a Dirichlet prior $K_i \sim P(K_i) = \mathbf{Dir}(\alpha_{i,0}, \ldots, \alpha_{i,|\mathcal{B}|-1})$. The prior is informed of the LDBA structure by setting $\alpha_{i,j} = 0$ if the LDBA does not allow transitions from $b_i$ to $b_j$. For the $m$ remaining non-zero Dirichlet parameters, we adopt a partially symmetric prior by setting them to a shared value $\frac{\alpha}{m}$, where $\alpha$ is a hyperparameter controlling the strength of the prior: large values of $\alpha$ induce slower convergence of the posterior distribution to the empirical transition kernel.

We remark that the choice of symmetry corresponds to the assumption that, at each step, the agent is capable of transitioning to each adjacent LDBA state with equal probability. In practice, the agent might actually take several steps in the underlying MDP in order to transition to any different LDBA state; moreover, some LDBA transitions can be substantially harder to achieve than others. Finally, for complex problems, naive exploration may not even result in observing the full set of possible transitions in a practical number of time steps. However, assuming that the agent is capable of transitioning under a max-entropy distribution allows reward signals to propagate to all states, thus resulting in informative estimates for the high-level value $\bar{V}_K$ and the intrinsic rewards shaping terms $R_{\text{intr}}$. For the illustrative example $T_0$, this is displayed in the bottom part of Figure 3.

Furthermore, while the initialization of $K$ is possibly unrealistic, the Bayesian framework provides a natural way to update it as an $N$-step trajectory $D = (b)_0^N$ is gathered in the product MDP:

$$P(K_i|D) \propto P(D|K_i)P(K_i) \tag{6}$$

for each $i \in [0, \ldots, |\mathcal{B}|]$. This update is tractable due to the choice of a conjugate Dirichlet prior for the categorical likelihood described by the transition kernel. As training progresses, a posterior distribution $P(K|D)$ can be updated with collected evidence. In practice, the updated posterior for state $b_i \in \mathcal{B}$ is simply $P(K_i|D) = \mathbf{Dir}(\alpha_{i,0} + c_{i,0}, \ldots, \alpha_{i,|\mathcal{B}|-1} + c_{i,|\mathcal{B}|-1})$, where $c_{i,j}$ is the count of LDBA transitions from state $b_i$ to state $b_j$ observed so far. Moreover, instead of computing high-level values for a specific transition kernel $K$ as in Equation 4, we can compute the expected value $\bar{V}$ under the posterior distribution of transition kernels $P(K|D)$ and thus of MRPs:

$$\bar{V} = \mathop{\mathbb{E}}_{K \sim P(K|D)}[\bar{V}_K] = \mathop{\mathbb{E}}_{K \sim P(K|D)}[(I - \gamma^\times \odot K)^{-1}]R^\times. \tag{7}$$

We remark that the maximization of $\bar{V}$ corresponds to the average-case MDP problem, while other procedures can be easily adapted to solve the Robust MDP (Xu & Mannor, 2010) or the Percentile Criterion MDP problems (Delage & Mannor, 2010). In practice, the expectation in Equation 7 can be estimated by sampling, with the special case of Thompson Sampling when a single sample is used. Our approach to estimating the transition kernel is ablated empirically in Appendix F; a study of the hyperparameter $\alpha$ controlling prior strength is in Appendix G.

### 3.3 Practical Algorithm

This section combines the elements outlined above into a practical scheme for distilling an intrinsic reward, which is reported in Algorithm 1. On top of the ability to sample trajectories from the product MDP and access to its LDBA, the algorithm only requires the specification of a prior distribution $P(K)$ over LDBA transition kernels, which is proposed in Section 3.2. The output is a policy $\pi$ optimizing an eventually discounted proxy to the likelihood of task satisfaction.

---

**Algorithm 1** DRL$^2$

---

**Input:** Product MDP $\mathcal{M}^\times$, prior distribution $P(K)$
**for** each iteration **do**
  Execute policy $\pi$ in $\mathcal{M}^\times$ to collect data $D$ for $N$ steps
  Update posterior $P(K|D)$ with evidence $D$ (Eq. 6)
  Compute high-level values $\bar{V}$ (Eq. 7)
  Sample training batch $B$ (either on- or off-policy)
  Add intrinsic reward to batch $B$ (Eq. 5)
  Train $\pi$ with $B$ through arbitrary RL algorithm
**end for**

---

## 4 Experiments

This section presents an empirical evaluation of the method by investigating the following questions:

- Can DRL$^2$ drive deep exploration in reinforcement learning from LTL specification?
- How does DRL$^2$ perform across different environments and specifications?
- Can DRL$^2$ be scaled to high-dimensional continuous settings?

As the method is designed to handle the full complexity of LTL, our evaluation considers several logic specifications, encompassing reach-avoidance (e.g., $Fa\&G(\neg b)$) and subtask sequences (e.g., $T_0 : GF(a\&XFc)\&G\neg b$). In order to focus on exploration, the suite of formulas allows easily scaling the number of LDBA states, or the minimum number of steps required in the underlying MDP to induce a transition in the LDBA. To investigate the final question, we perform an evaluation in both *tabular* and *continuous* domains. While the former avoids confounding effects arising from function approximation, the latter stresses the ability to scale to complex underlying environments. This also evaluates the versatility of DRL$^2$, as it is in practice coupled with tabular Q-learning (Watkins & Dayan, 1992) and Soft Actor Critic (Haarnoja et al., 2018a), respectively. In the first case, the environment involves navigation in a 2D gridworld, while in the latter we evaluate dexterous manipulation with a simulated Fetch robotic arm and locomotion of a 12DoF quadruped robot and a 6DoF HalfCheetah. A detailed description of the benchmark environments and specifications is provided in Appendix M; code is available at github.com/marbaga/drl2.

**Baselines** We compare DRL$^2$ to:

1. a counterfactual experience replay scheme (LCER (Voloshin et al., 2023)),
2. a novel count-based baseline [4], that relies on inverse square root visitation counts of LDBA states to compute a potential function for reward shaping (Tang et al., 2017). If $n(b)$ is the visitation count of LDBA state $b \in \mathcal{B}$ from the beginning of training, the intrinsic reward for the high-level transition $b \to b'$ is simply computed as $R_{\text{intr}}(b, b') = \gamma^\times(b')\frac{1}{\sqrt{n(b')}} - \frac{1}{\sqrt{n(b)}}$.
3. the underlying learning algorithm with no additional exploration bonuses; this corresponds to simply setting $R_{\text{intr}}(b, b') = 0$ in DRL$^2$.

All shared hyperparameters are set to the ones maximizing the performance of the *No exploration* baseline; the prior strength $\alpha$ is the main hyperparameter introduced by DRL$^2$, and is simply set to $10^3$ in all tabular experiments, and $10^5$ in all continuous experiments. For a thorough discussion on hyperparameters, we refer to Appendix M. We note that other promising approaches for exploration in the LTL domain exist, but they rely on meta-learned components or ad-hoc training regimes (Qiu et al., 2023; Wang et al., 2023) and are thus not suitable for a fair comparison.

---

[4]This baseline is, to the best of our knowledge, novel, as the application of standard RL exploration ideas to the MRP/MDP derived from the LDBA has not been thoroughly investigated. We choose a count-based approach as the MRP is discrete, and does not require hashing or continuous generalizations Burda et al. (2019), while a straightforward application of reward-based optimistic methods is not possible, as the reward of the MRP is known.

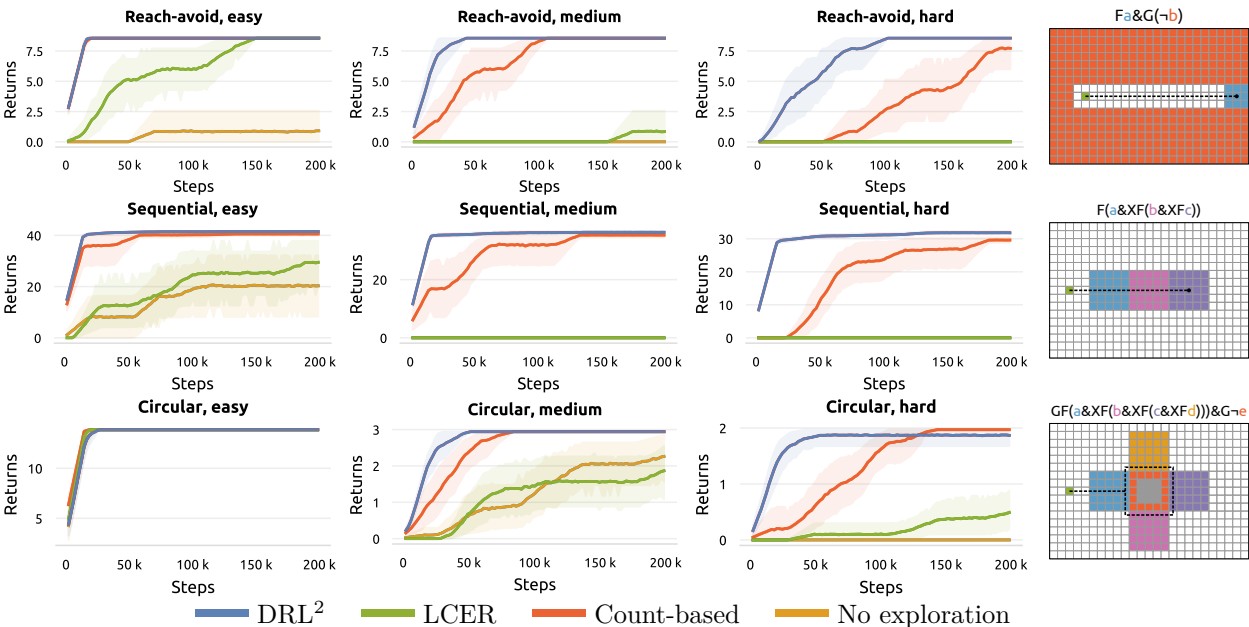

Figure 4: Return for Q-learning under eventual discounting. The three rows display reach-avoidance, sequential and circular tasks, as illustrated on the right with optimal policies. Each atomic proposition evaluates to $\top$ in cells of matching color; difficulty increases from left to right. Further details are available in Appendix M. DRL$^2$ is able to drive exploration when naive exploration is insufficient.

## 4.1 Tabular Setting

We first evaluate the exploration performance of DRL$^2$ by coupling it with tabular Q-learning (Watkins & Dayan, 1992) when operating over discrete state and action spaces in the standard online episodic setting (Sutton & Barto, 2018). The evaluation environment is a deterministic 2D gridworld, in which the agent can move in each of the four cardinal directions by one unit at each timestep. The first row of Figure 4 evaluates a reach-avoidance task $G(a\&\neg b)$, in which the agent needs to avoid a large area that covers all but a corridor. The goal area is at the end of said corridor; and the difficulty of the task increases with the length of the corridor. In this case the benefit of DRL$^2$ is evident, as it returns a negative reward whenever the agent leaves the corridor, and the LDBA thus transitions to a sink state. The agent can therefore direct its exploration towards a fraction of the state space, resulting in improved sample efficiency. On the other hand, count-based shaping can only penalize transitions to the sink state once it has been visited enough times. While LCER has the advantage of potentially ignoring failures by hallucinating counterfactual LDBA states during training, it cannot discourage exploration of the forbidden area. We remark that LCER remains a strong method significant when exploration of the underlying MDP is not necessary; a detailed discussion is provided in Appendix E.

The second and third row of Figure 4 evaluate sequential tasks. In both, the agent navigates a room with several zones. In the first case, the agent must reach a sequence of zones in a given order; in the second one, this must be repeated indefinitely while also avoiding the center of the room. While the standard reward under eventual discounting would be zero until the last zone in the sequence is reached, DRL$^2$ provides an informative reward at each LDBA transition, thus informing the agent to direct exploration towards promising directions. Therefore, as number and size of the zones grows (to the right), DRL$^2$ results in more efficient exploration by encouraging LDBA transitions towards the accepting state. This encouragement is instead only dependent on visitation counts for the count-based baseline and absent in LCER. We additionally evaluate variations of these tasks in Appendix D.

## 4.2 Continuous Setting

After verifying the effectiveness of DRL$^2$ in interpretable settings, this section investigates if the exploration signal can be scaled to high-dimensional, long-horizon environments requiring the application of deep reinforcement learning algorithms. For simplicity, we evaluate its application to Soft Actor Critic (SAC)

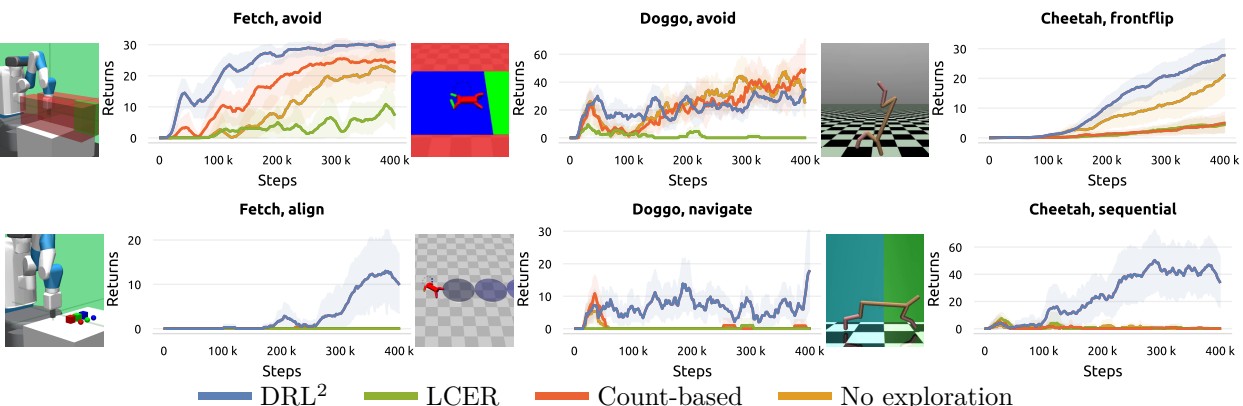

Figure 5: Return for SAC under eventual discounting on Fetch (left), Doggo (center) and HalfCheetah (right), as shown in renderings. DRL$^2$ confirms its ability to drive exploration in complex tasks when coupled with deep reinforcement learning.

(Haarnoja et al., 2018a), which stands as a fundamental building block for many algorithms (Eysenbach et al., 2019; Haarnoja et al., 2018b; Ibarz et al., 2021; Kumar et al., 2020). On the top of Figure 5 a simulated Fetch robotic arm (de Lazcano et al., 2023) is evaluated on two tasks, namely (i) moving its gripper to a specific location while avoiding lateral movements and (ii) gradually producing an horizontal alignment of three cubes. In the middle, an HalfCheetah receives specifications encoding, respectively, finite sequences of positions and infinite sequences of angles for its center of mass, resulting in precise horizontal locomotion, and in front flipping indefinitely. On the bottom, a 12DoF simulated quadruped robot (Ray et al., 2019) is tasked with (i) fully traversing a narrow corridor, or with (ii) navigating through two zones in sequence. As in the tabular case, the four tasks can all be described through LTL formulas encoding reach-avoidance and sequential behavior. They are reported among further details on the environments in Appendix M.

As expected, in this setting the evaluation is significantly noisier and partially constrained by the learning algorithm. Nevertheless, we observe that DRL$^2$ is competitive with the stronger baselines in simpler problems, and likely to outperform them when exploration in the underlying MDP becomes more challenging. Interestingly, we observe that counterfactual experience replay (LCER) is less effective in this setting. We remark that LCER can generate unfeasible states for the product MDP, which can be harmful when the agent has the ability to interpolate between training samples.

## 5    Related Work

**Logic for Task Specification in RL**    LTL is among several logic languages for task specification in reinforcement learning: for instance, numerous related works have investigated Reward Machines (Camacho et al., 2019; Icarte et al., 2018; 2022; Vaezipoor et al., 2021). While Reward Machines can handle regular expressions, LTL is designed to address a superset, namely $\omega$-expressions. As a consequence, LTL gains the ability to reason about infinite sequences and tasks including liveness and stability.

For this reason, several works, including our own, have investigated LTL for task specification. Early attempts focused on reconciling logic and policy optimization through the definition of a product MDP and the design of a reward signal encouraging task satisfaction (Bozkurt et al., 2020; Camacho et al., 2017; 2019; Hasanbeig et al., 2018; 2020; Kantaros, 2022; Li et al., 2017). Several works relied on heuristic reward shaping schemes (Li et al., 2017), which gradually evolved towards dynamic or eventual rewarding and discounting schemes (Cai et al., 2021; De Giacomo et al., 2020a;b; Hasanbeig et al., 2018; 2020; Voloshin et al., 2023). This led to the development of principled approaches, proposing schemes that provably and directly optimize a lower bound on the likelihood of formula satisfaction (Shao & Kwiatkowska, 2023; Voloshin et al., 2023).

As for the optimization of such schemes, while linear programming is sufficient for known dynamics and finite action spaces, Q-learning approaches have been adopted when a model of the environment is not available (Bozkurt et al., 2020; Cai et al., 2021; Hasanbeig et al., 2018). A fundamental restriction of this approaches was lifted by adopting actor-critic (Hasanbeig et al., 2020) or policy gradient (Voloshin et al.,

2023) methods suited for continuous action spaces. Nevertheless, in this case, empirical demonstration of LTL-guided reinforcement learning have remained mostly constrained to low-dimensional setting.

This work builds on top of these efforts, and adopts established rewarding and discounting schemes. We focus, however, on the specific problem of exploration, and propose a method to densify the reward signal. Furthermore, leveraging modern actor-critic methods, we extend our empirical evaluation to high-dimensional, complex environments.

**Exploration in Logic-driven Reinforcement Learning**  A significant effort has been targeted at addressing the sparsity arising from existing rewarding schemes (Hasanbeig et al., 2018; Voloshin et al., 2023), involving the use of heuristics (Li et al., 2017), an accepting frontier function (Hasanbeig et al., 2018; 2020), bonuses for first visitations (Cai et al., 2021), or the knowledge of additional information such as annotated maps (Xiong et al., 2023). Recently, Shah et al. (2024) also propose a heuristic scheme for shaping constraints expressed in LTL. This scheme is partially aligned with the shaping signal that DRL$^2$ naturally recovers, but only considers accepting paths and is restricted to on-policy methods. Directly within the LTL literature, another line of work leverages the discrete structure of automata and learns hierarchical (Hasanbeig et al., 2020; Icarte et al., 2022; Jothimurugan et al., 2021), goal-conditioned (Qiu et al., 2023) or modular (Cai et al., 2021) policies. While considerably improving sample efficiently, the approaches listed so far are known to potentially introduce suboptimality (Icarte et al., 2022). In contrast, DRL$^2$ recovers the optimal policy under mild assumptions, as described in Appendix A.

In the parallel direction of Reward Machines, several works investigate reward shaping for reinforcement learning from logic specification. Camacho et al. (2017) consider potential functions $\Phi(s, b)$ of both the MDP and automaton states; among those, they investigate counting the minimum number of steps to reach an accepting automaton state, which is equivalent to computing values under a specific automaton transition kernel. Another work (Icarte et al., 2022) explores an extension, which is also fundamentally related to our method: the authors investigate value iteration on an MDP synthetised from the automaton to compute a potential function for reward shaping, thus leveraging the semantics of the task. While motivated by a similar intuition to ours, a naive application of this method would not be viable in an eventually discounted setting, which involves $\omega$-regular problems, as we report in Appendix F. By adopting an LDBA instead of a DFA, and more importantly by integrating an eventual discounting scheme, our method is able to avoid myopic behavior while handling the full complexity of LTL. Futhermore, DRL$^2$ does not require assumptions on controllability by defining a Markov reward process instead. Values estimated in the MRP are thus softer than the optimal ones in a corresponding MDP, and we find them to be more informative for exploration. We validate this discussion empirically, by evaluating optimal transition kernels under eventual discounting in Appendix F. By updating its distribution over MRP transition kernels, it is moreover capable of providing an adaptive and informative reward signal, rather than a static one derived from a hard assumption.

Beyond reward shaping, some works leverage control over the automaton to produce synthetic experience by relabeling the LDBA states of collected transitions. Such schemes have been proposed in the context of Reward Machines (Icarte et al., 2022) and LTL (Voloshin et al., 2023); as they can produce off-policy samples, they cannot be naively applied to on-policy methods, as discussed in Appendix E. Other relabeling techniques only target the initial LDBA state (Wang et al., 2020) or focus on optimality guarantees (Shao & Kwiatkowska, 2023). These methods are essentially orthogonal to DRL$^2$ (see Appendix E), and may be combined. Finally, we note that previous work have also led to strong exploration on unseen tasks, as the result of a meta-learning phase when a distribution over MDPs can be accessed (León et al., 2022; Vaezipoor et al., 2021).

To the best of our knowledge, our approach is novel in its adaptive and informed estimation of the high-level Markov reward process, resulting in a directed exploration method that retains optimality.

**Directed Exploration in Reinforcement Learning**  In this work, we use the word *directed* with a specific meaning, as DRL$^2$ *directs* the agent towards accepting LDBA states, by rewarding LDBA transitions towards them. In contrast, count-based methods promote transitions to any unvisited LDBA state, and may thus be considered undirected. Nonetheless, this idea has parallels in non-logic-conditioned reinforcement learning, as a direction can be given with respect to entities beyond accepting states. In this context, *directed*

has been used in conjunction to the Bayesian notion of information, in order to promote exploration of states that are maximally informative of some quantity (e.g., a reward (Lindner et al., 2021), a value estimate (Nikolov et al., 2019), or learned dynamics (Guo & Brunskill, 2019)).

Another way to provide direction or guidance is in the context of action-masking methods (Stolz et al., 2024; Theile et al., 2024), which pose hard constraints on actions that can be selected by the agent. $DRL^2$ provides a milder feedback through an intrinsic reward term, and does not explicitly prevent actions from being taken. Finally, many works have explored goal-conditioned RL as a technique to guide and improve exploration (Eysenbach et al., 2022), with several works at the intersection with LTL (Qiu et al., 2023; Yalcinkaya et al., 2024). Similarly to goal-conditioned RL, our work extracts a reward from a specific task (in our case, a logic-formula); however, this task is unique, and we do not consider the standard multi-goal/universal value function formulation (Schaul et al., 2015) that is at the core of goal-conditioned RL.

## 6  Discussion

This work proposes $DRL^2$, an exploration method for reinforcement learning from LTL specifications. By casting the LDBA encoding the specification as a Markov reward process, we enable a form of high-level value estimation, which can produce value estimates for each LDBA state. These values can be leveraged as a potential function for reward shaping and combined with an informed Bayesian estimate of the LDBA transition kernel to ensure an informative training signal. As a result, $DRL^2$ accelerates learning when non-trivial exploration of the underlying MDP is necessary for reaching an accepting LDBA state, as is often the case for complex specifications.

**Limitations**  $DRL^2$ is generally capable of improving exploration in LTL-conditioned RL; however, we do not expect it to be useful for all tasks, and we suggest four such instances. First, while $DRL^2$ reduces sparsity in reward signal distilled from LTL specifications, the shaped reward is not entirely dense. Intuitively, instead of receiving a non-zero reward when an accepting state is visited, the agent receives it at every LDBA transition. However, if naive exploration is insufficient to induce an LDBA transition, exploration remains challenging. For this reason, we expect our method to be insufficient in environments with extremely long horizons and rare LDBA transitions. Second, in tasks in which each MDP state can be associated with many LDBA states, or presenting $\epsilon$-transitions, a replay method such as LCER is often sufficient, and, while it can be seamlessly combined, $DRL^2$ may not provide significant improvements (see Appendix E). Third, $DRL^2$ relies on an symmetric prior over LDBA transitions, which is in general effective, but could be misleading for specific tasks and MDP morphologies. Fourth, as several other exploration schemes, we note that $DRL^2$ introduces a slight non-stationarity in the reward signal, which needs to be addressed by the underlying learning algorithm and might be problematic in very stochastic environments, especially for off-policy algorithms. Nevertheless, this is a common issue of intrinsic exploration methods, which is in practice often neglectable Burda et al. (2019). Finally, and perhaps foremost, $DRL^2$ relies on the availability of a *correct* LDBA, and a ground-truth, accurate labeling function to infer AP evaluations from states. While this assumption is shared by the bulk of works in LTL-conditioned RL, real-world deployment of such algorithms would require algorithms that are robust to partial observability and noise, and capable of jointly inferring the labeling function from data, which we highlight as an important research direction.

**Outlook**  This work opens up exciting future directions, including a formal analysis of which reward shaping terms would not only grant consistency, but also maximize sample efficiency. Moreover, as $DRL^2$ leverages the LDBA structure, its effectiveness is dependent on it. Its applicability can thus further benefit by LDBA construction algorithms that do not return an arbitrary LDBA in its equivalence class, but rather the one which is most suitable for guiding a reinforcement learning agent.

**Conclusion**  This work addresses the fundamental problem of sparsity for complex LTL tasks by directly leveraging their structure. Thus, we believe that the evidence provided in this work further supports the effectiveness of reinforcement learning as a paradigm for extracting a controller from a logic specification.

**Acknowledgments**

We thank Nico Gürtler for precious discussions throughout the project, and the anonymous reviewers for their valuable feedback. Marco Bagatella is supported by the Max Planck ETH Center for Learning Systems. This project has received funding from the Swiss National Science Foundation under NCCR Automation, grant agreement 51NF40 180545. Georg Martius is a member of the Machine Learning Cluster of Excellence, EXC number 2064/1 – Project number 390727645. We acknowledge the support from the German Federal Ministry of Education and Research (BMBF) through the Tübingen AI Center (FKZ: 01IS18039B).

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

# A  Proof of Proposition 3.1 and Further Analysis

This section presents a proof for Proposition 3.1 in Section 3, which we also report in its entirety for ease of reference, and further discusses how the intrinsic reward is informed by the task.

**Proposition A.1.** *(Consistency) Let us consider the product MDP $\mathcal{M}^\times = (\mathcal{S}^\times, \mathcal{A}^\times, \mathcal{P}^\times, R^\times, \gamma^\times, \mu_0^\times)$ and its modification $\widetilde{\mathcal{M}}^\times = (\mathcal{S}^\times, \mathcal{A}^\times, \mathcal{P}^\times, R^\times + R_{intr}, \gamma^\times, \mu_0^\times)$. Under eventual discounting, any optimal policy in $\widetilde{\mathcal{M}}^\times$ for which $\Gamma_t \overset{t \to \infty}{\to} 0$ is also optimal in $\mathcal{M}^\times$.*

The proof consists of a simple extension of known results on potential-based reward shaping (Ng et al., 1999) to account for eventual discounting. Let us consider an arbitrary state-action pair $(s_0, a_0) \in \mathcal{S}^\times \times \mathcal{A}^\times$ and the optimal Q-function in $\widetilde{\mathcal{M}}^\times$ under eventual discounting :

$$\tilde{Q}_\star^\gamma(s_0, a_0) = \underset{\pi^\star, P^\times}{\mathbb{E}} \left[ \sum_{t=0}^\infty \Gamma^t (R^\times(s_t, a_t, s_{t+1}) + R_{\text{intr}}(s_t, a_t, s_{t+1})) \right] \tag{8}$$

$$= \underset{\pi^\star, P^\times}{\mathbb{E}} \left[ \sum_{t=0}^\infty \Gamma^t R^\times(s_t, a_t, s_{t+1}) + \sum_{t=0}^\infty \Gamma^t R_{\text{intr}}(s_t, a_t, s_{t+1})) \right] \tag{9}$$

$$= Q_\star^\gamma(s_0, a_0) + \underset{\pi^\star, P^\times}{\mathbb{E}} \left[ \sum_{t=0}^\infty \Gamma^t R_{\text{intr}}(s_t, a_t, s_{t+1})) \right] \tag{10}$$

$$= Q_\star^\gamma(s_0, a_0) + \underset{\pi^\star, P^\times}{\mathbb{E}} \left[ \sum_{t=0}^\infty \Gamma^t (\gamma^\times(s_{t+1}) \bar{V}(b_{t+1}) - \bar{V}(b_t)) \right] \tag{11}$$

$$= Q_\star^\gamma(s_0, a_0) + \underset{\pi^\star, P^\times}{\mathbb{E}} \left[ \sum_{t=0}^\infty \Gamma^t \gamma^\times(s_{t+1}) \bar{V}(b_{t+1}) - \sum_{t=0}^\infty \Gamma^t \bar{V}(b_t)) \right] \tag{12}$$

$$= Q_\star^\gamma(s_0, a_0) + \lim_{t \to \infty} \underset{\pi^\star, P^\times}{\mathbb{E}} \left[ \Gamma^t \bar{V}(b_t) - \gamma^\times(s_0) \bar{V}(b_0) \right] \tag{13}$$

$$= Q_\star^\gamma(s_0, a_0) + \lim_{t \to \infty} \underset{\pi^\star, P^\times}{\mathbb{E}} \left[ \Gamma^t \bar{V}(b_t) \right] - \gamma^\times(s_0) \bar{V}(b_0) \tag{14}$$

where $Q_\star^\gamma$ is the optimal Q-function in $\mathcal{M}^\times$ under eventual discounting. If the assumption holds, and $\Gamma_t \overset{t \to \infty}{\to} 0$, then the second term fades, and we have

$$\tilde{Q}_\star^\gamma(s_0, a_0) = Q_\star^\gamma(s_0, a_0) - \gamma^\times(s_0) \bar{V}(b_0) \tag{15}$$

The difference between the two Q-functions $\tilde{Q}_\star^\gamma$ and $Q_\star^\gamma$ is therefore constant at any given state. Thus,

$$\tilde{\pi}^\star(s) = \underset{a \in \mathcal{A}^\times}{\operatorname{argmax}} \tilde{Q}_\star^\gamma(s, a) = \underset{a \in \mathcal{A}^\times}{\operatorname{argmax}} Q_\star^\gamma(s, a) = \pi^\star(s). \quad \square \tag{16}$$

As stated above, this invariance relies on the assumption that the product of discounts converges to zero, which would happen in case an accepting state is visited infinitely often [5]. Intuitively, once a good policy is found, the contribution of the intrinsic reward fades, and there is no incentive for optimization algorithms to update the policy away from its optimum. While this assumption often holds for the optimal policy (e.g., in deterministic environments), there are cases in which this may not be true (e.g., in the stochastic case, if the LDBA features a sink state). Nevertheless, we note that the assumption holds for optimal policies in tasks considered in this paper, and in recent literature (Voloshin et al., 2023).

---

[5]Formally, convergence to zero only happens if the agent visits an accepting state infinitely many times over an infinite horizon, which occurs if and only if the policy satisfies the formula. In practice, $\Gamma_t$ decays exponentially fast as the number of visits until $t$ increases: if $\gamma = 0.99$ and an accepting state has been visited, say, 1000 times, then $\Gamma_t \approx 10^{-4}$.

On the other hand, in cases in which the assumption does not hold (e.g., as often happens for a random initialized policy), the intrinsic reward can in principle still bias the policy and drive exploration well. For instance, let us assume that the likelihood of visiting an accepting LDBA state under the current policy $\pi$ is exactly zero. This implies $R^\times(\cdot) = 0$, $\gamma^\times(\cdot) = 1$ and $\Gamma_i = \prod_{t=0}^{i-1} \gamma^\times(b_t) = 1$. In this case, the value under eventual discounting for an arbitrary state $s_0 \in \mathcal{S}^\times$ is

$$\tilde{V}_\pi^\gamma(s_0, a_0) = \mathop{\mathbb{E}}_{\pi^\star, P^\times} \left[ \sum_{t=0}^\infty \Gamma^t (R^\times(s_t, a_t, s_{t+1}) + R_{\text{intr}}(s_t, a_t, s_{t+1})) \right] \tag{17}$$

$$= \mathop{\mathbb{E}}_{\pi^\star, P^\times} \left[ \sum_{t=0}^\infty R_{\text{intr}}(s_t, a_t, s_{t+1}) ) \right] \tag{18}$$

$$= \mathop{\mathbb{E}}_{\pi^\star, P^\times} \left[ \sum_{t=0}^\infty \gamma^\times(s_{t+1}) \bar{V}(b_{t+1}) - \bar{V}(b_t) \right] \tag{19}$$

$$= \lim_{t \to \infty} \mathop{\mathbb{E}}_{\pi^\star, P^\times} \left[ \bar{V}(b_t) \right] - \bar{V}(b_0). \tag{20}$$

Thus, a policy optimizing the value function under eventual discounting with the intrinsic reward provided by DRL$^2$ also maximizes the likelihood of eventually reaching the LDBA state with maximum high-level value $\bar{V}$. It is easy to show that this corresponds to the accepting state $b^\star$ in the LDBA as the high-level value of any LDBA state $b \in \mathcal{B}$ is $\bar{V}(b) = (1 - P_\pi(\neg b^\star | b)) \bar{V}(b^\star) \leq \bar{V}(b^\star)$, where $P_\pi(\neg b^\star | b)$ is the likelihood of never reaching the accepting state under the current policy.

We can thus conclude that a policy trained with DRL$^2$ preserves its asymptotic optimality, while also benefiting from an informative exploration signal during early stages of training.

## B  Connection Between MRP and Product MDP

This section discusses the relationship between product MDP (introduced in Section 2) and MRP (described in Section 3) in further detail. Let us consider a trajectory in the product MDP $\tau = ((s_0, b_0), (s_1, b_1), \dots)$. Due to the construction of the MRP, the return of the projection of $\tau$ to the MRP state space (in this case, $(b_0, b_1, \dots)$) is the same as the return of $\tau$ in the product MDP, evaluated according to $R^\times$ and $\gamma^\times$. A similar argument can be made for a policy $\pi$ given a fixed starting state $(s_0, b_0)$. If the MRP transition kernel $K$ is a projection of the transition kernel induced by $\pi$ in the product MDP, then the value $V_K(b_0)$ computed in the MRP matches the value computed in the product MDP $V^\pi((s_0, b_0))$. Nonetheless, we remark that the MRP can be seen as a high-level approximation of the product MDP, and as such it loses information. Thus, it may not be straightforward to leverage the MRP to accurately compute values for arbitrary MDP states, to the best of our knowledge.

## C  Addition of a Virtual Sink State

As described in Section 3.1, a design choice in DRL$^2$ involves the construction of a virtual sink state for LDBAs that do not feature one, where a sink state can be defined as an LDBA state $b_s \in \mathcal{B}$ such that $P^\mathcal{B}(b_s, \cdot) = b_s$.

Let us consider an LDBA featuring a path from each state to an accepting state. Moreover, let us consider a transition kernel $K$ with full support over reachable LDBA states (as is the case for likely samples under the prior proposed in Section 3.2). In this case, the values computed under eventual discounting by solving Equation 7 would be uniform: $\bar{V}_K = \frac{1}{1-\gamma}$. As a consequence, intrinsic rewards computed through Equation 5 would be constantly zero until an accepting state is visited and thus uninformative. This is a natural consequence of the fact that, if irreversible failure is not possible, any policy inducing a transition kernel with a good enough support will satisfy the specification, considering an infinite horizon.

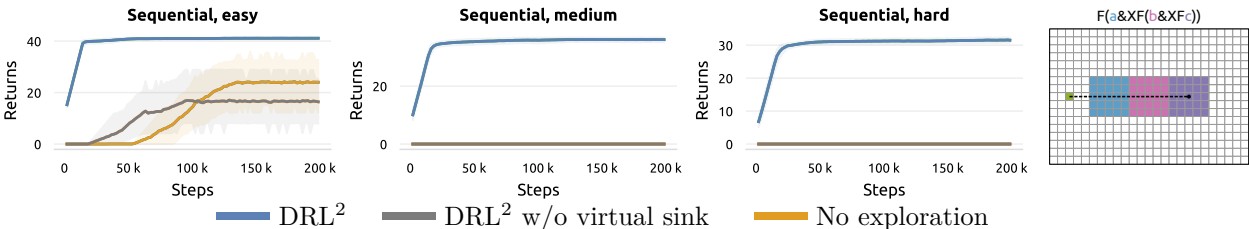

Figure 6: Ablation of the addition of a virtual sink for DRL². When a sink state is not featured nor added (e.g. in sequential tasks), the performance of the No-exploration baseline is recovered.

In order to provide an informative learning signal, DRL² augments LDBAs that do not feature a sink state with a virtual one and assumes that it can be reached from any other LDBA state. We note that this is equivalent with associating each LDBA state with a discount factor equal to the likelihood of transitioning to the virtual sink. While this can introduce myopic behavior (Voloshin et al., 2023), we remark that DRL² also learns and updates its estimate of the LDBA transition kernel. As any virtual sink state cannot in practice be reached, the likelihood of transitioning to it from each visited LDBA state converges to zero as the number of environment steps increases. Thus, as training progresses, any virtual sink is effectively removed, together with any degree of myopia introduced.

Among the tasks considered in Figure 4, MRPs for sequential tasks do not naturally feature a sink state. To support the discussion, we additionally provide empirical evidence on the performance of DRL² in these tasks when a virtual sink is not added. We note that performance on other tasks would not be affected, as they naturally feature a sink state.

In Figure 6 we observe that DRL² without a virtual sink state recovers the performance of the No-exploration baseline, as expected. Experimental settings are the same as those outlined in Section 4.

# D  Additional Tabular Results

**UMaze**   Optimal policies for the reach-avoidance tasks from Figure 4 (first row) would also solve a shortest-path reaching problem in the underlying MDP. In other words, a linear path to the goal does not violate the constraints. We now evaluate a variation of the original task, by reshaping the safe area of the reach-avoid task into a U-shape. In this case, the shortest path to the goal violates the avoidance constraints. Fortunately, changing the shape of the avoidance zone only results in a local change to the labeling function $\mathcal{F}$. DRL², as well as other baselines, can handle arbitrary relabeling functions; thus these methods can adapt to arbitrary environment layouts.

Results are reported in 7. for all task variants (easy, medium and hard). In general, as the path length has now increased, performance globally decreases accordingly. Nevertheless, the overall relative performance of each methods is consistent with the original version of the task. We note that LCER (Voloshin et al., 2023) now outperforms the count-based baseline, as it can effectively ignore previous visits to the zone to avoid. In the extreme case in which the length of the U-maze is very large, but the starting position is very close to the goal if the avoidance constraint is ignored, we expect LCER to perform even better.

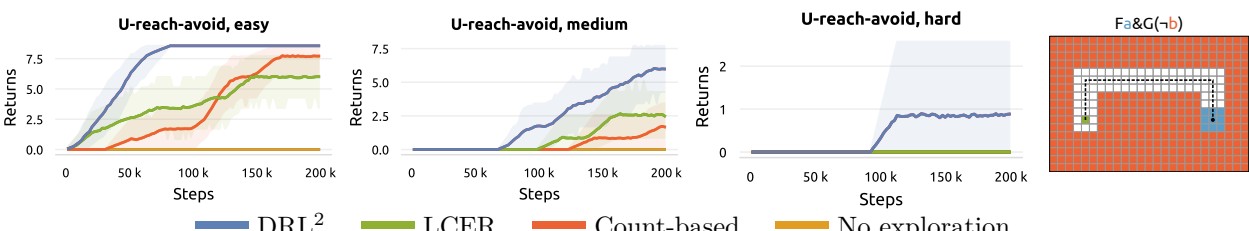

Figure 7: Alternative evaluation of reach-avoid task involving a conflict between reaching objectives and avoidance constraints. Results are consisted with those in Figure 4.

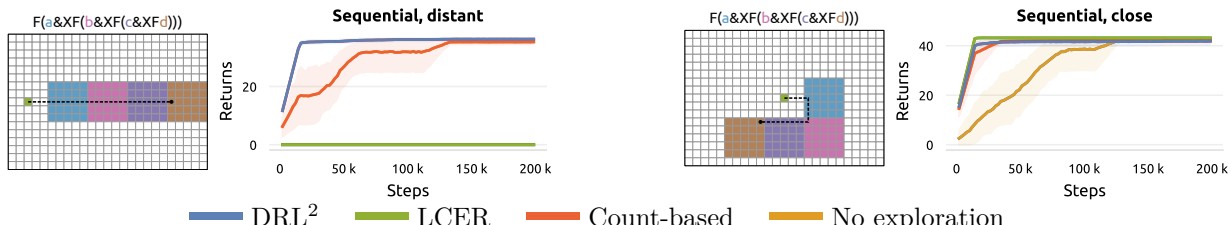

Figure 8: Alternative evaluation of a sequential task in the shape of a maze. Results are consistent with those in Figure 4.

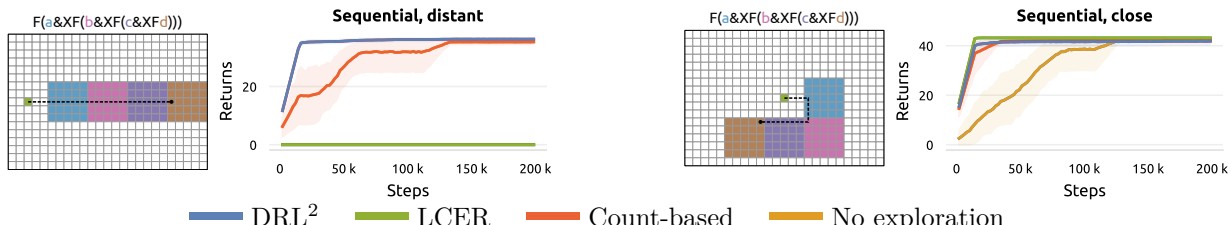

Figure 9: Comparison of DRL$^2$, LCER and additional baselines in the Sequential task from Figure 4 (left) and an additional variation (right). LCER performs very well when significant exploration of the underlying MDP is not required, as is the case for the example on the right. To the left of each set of training curve, the task is represented, and an optimal policy is visualized.

**Sequential maze** We additionally evaluate an alternative layout of the sequential task, which considers a greater number of APs, and in which the sequence of zones to visit leads to the exit of a maze with several intersections (see Figure 8). We scale the difficulty by increasing the areas of each zone. As the reward signal synthesized by DRL$^2$ mostly depends on the structure of the LDBA, we observe consistent results with the sequential tasks in Figure 4.

# E  Comparison to LTL-guided Counterfactual Experience Replay (LCER)

The environments and tasks presented in Section 4 require significant exploration in the underlying MDP and stress the ability to leverage the semantics of LDBA states to accelerate this process.

LTL-guided Counterfactual Experience Replay (Voloshin et al., 2022) takes an orthogonal approach to the problem of exploration when learning from LTL formulas and represents a very strong baseline in specific environments. In particular, LCER does not leverage the semantics of the LDBA, however it generates synthetic experience by fully controlling its dynamics. Transitions $((s, b), a, (s', b'))$ with $\{(s, b), (s', b')\} \subseteq \mathcal{S}^\times$ and $a \in \mathcal{A}^\times$ are relabeled by uniformly resampling the initial LDBA state and simulating the LDBA transition caused by action $a$, resulting in a new transition $((s, \hat{b}), a, (s', \hat{b}'))$.

Generated transitions respect the transition kernel of the product MDP and are thus valid training samples. However, the method does not guarantee that relabeled product states $(s, \hat{b})$ are actually feasible, as the set of reachable product states might only consist of a manifold of the full Cartesian product of MDP and LDBA state spaces. For instance, let us consider the illustrative task $T_0$ as displayed in Figure 2: at any step, if the agent is in the red field, the LDBA would necessarily be in its sink state and in no other state. Any relabeling of this particular state would thus not be relevant. Furthermore, as a form of state relabeling, LCER cannot generate on-policy data, as the on-policy action sampled in $(s, \hat{b})$ might be different than the original action sampled in $(s, b)$ (Rauber et al., 2019).

Nevertheless, LCER remains a very strong exploration method in environments in which each MDP state can be associated with several LDBA states; furthermore, in the presence of sink LDBA states, LCER can hallucinate their avoidance. In this merit, we report an additional experiment in Figure 9, evaluating DRL$^2$, LCER and baselines in a modified version of the Sequential environment from Figure 4. In its original version

Figure 10: Ablation of kernel initialization and estimation methods. Results suggest that an informed symmetric prior, as used by $DRL^2$, performs reliably.

(on the left), the accepting LDBA state can only be visited once the distant final zone (in bright purple) is reached. In this case, LCER is not effective. However, if the zones are rearranged in a way such that they can be visited without excessive exploration in the underlying MDP (e.g., by placing them closer to the starting MDP state, as done on the right), the performance of LCER largely improves, fully bridging the gap with $DRL^2$. We note that the modified version of the task requires fewer steps to reach an accepting state from the starting one; the baseline without exploration incentives also solves the task, albeit less efficiently. For evaluations on additional variations of tabular tasks, we refer the reader to Appendix D and I.

LCER and $DRL^2$ thus address different issues in exploration from logic formulas. Fortunately, the two methods are orthogonal and can be freely combined.

# F    Kernel Estimation Ablation

Proposition 3.1 states that, under specific assumptions, optimal policies are invariant to the proposed reward shaping. In fact, Proposition 3.1 holds for any choice of transition kernel $K$. In practice, the choice of kernel still impacts the quality of exploration and sample efficiency. We ablate this choice by comparing the effectiveness of kernels sampled by our methods with several others, to assert what constitutes a "good" kernel for exploration.

We remark that $DRL^2$ averages its values over transition kernels sampled from a posterior distribution to a symmetric Dirichlet prior. We compare this choice to three additional ones:

- kernels sampled from a partially informed prior, which differs from the one used in $DRL^2$ due to its lack of knowledge over the connectivity of the MRP/LDBA, except for its sink state. It essentially assumes that a transition between any LDBA state pair is always possible.

- a transition kernel obtained using value iteration (VI), and that therefore represents the optimal policy under assumption of full controllability of an high level MDP. It represents a naive implementation of the reward shaping introduced in Icarte et al. (2022). This kernel has the lowest entropy possible, as value iteration outputs a deterministic policy.

- an empirical kernel, which simply counts LDBA transitions performed by the policy in the product MDP. This kernel does not leverage the LDBA structure, and is visualized in Figure 3.

These ablations are evaluated in the medium variants of the three tabular tasks from Figure 4 to highlight performance differences; all experimental parameters are unchanged.

Results are presented in Figure 10. We observe that a partially informed prior can recover part of $DRL^2$'s performance, but suffers as the size of the LDBA increases (Sequential, medium). The transition kernel obtained through VI performs well in some tasks, and less in other: since the optimal policy under eventual discounting assigns similar values to non-sink LDBA states, reward shaping becomes less informative. Finally, choosing the empirical kernel generally leads to low performance, as discussed in Figure 3.

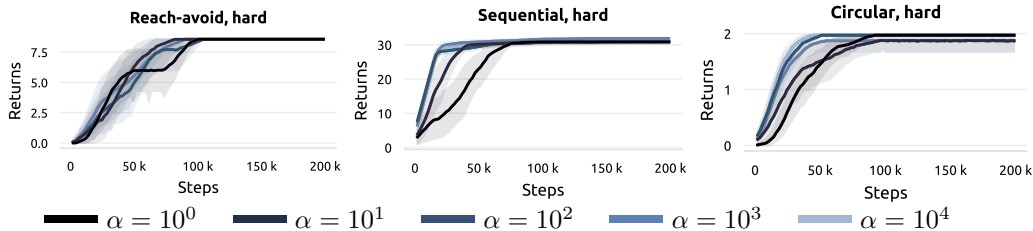

Figure 11: Ablation of prior strength hyperparameter $\alpha$.

# G    Prior Strength Ablation

This Section performs an empirical evaluation of the effect of the hyperparameter $\alpha$, which controls the strength of the prior over MRP transition kernels in $\text{DRL}^2$. We ablate the default choice ($\alpha = 1000$) in the hard variant of tasks from the tabular evaluation in Figure 4. Overall, in Figure 11 we observe that $\text{DRL}^2$ is fairly robust to this hyperparameter. Performance is generally lower for weak priors ($\alpha < 100$), which can be significantly affected by poor trajectories collected by the initial policy. As the prior is informed by the LDBA structure, we find that it generally provides good guidance even for stronger priors. We note that, in case the prior is misinformed, a weaker prior would indeed be favorable.

# H    Product MDP Ablation: States and Rewards

The product MDP formulation that our method builds upon (Voloshin et al., 2023) can be used to formally connect eventually discounted returns and likelihoods of formula satisfaction, but strictly requires a binary $1/0$ reward signal. Moreover, the agent state space is defined as the product between LDBA and MDP states. In this section, we empirically evaluate trends that arise if these two properties are violated.

**States**    In principle, the agent needs access to both high-level LDBA states (tracking formula satisfaction) and low-level MDP states, tracking fine-grained details. While this is necessary to provably recover an optimal policy for arbitrary MDPs and formulas, it also causes a blow-up of the state space, which makes exploration more difficult. As rewards are only defined as a function of the LDBA states, one might be tempted to directly learn a policy (and value function) from LDBA states alone $\pi : \mathcal{B} \to \mathcal{A}^\times$. This decision has the large advantage of operating over a much smaller state space, and the disadvantage of losing information about low-level MDP information. This is evident in the empirical evaluation we present for tabular tasks in Figure 12 (see curves for *LDBA-only*. Across all tasks, the optimal value function is not *invariant* with respect to the MDP state; thus, value estimates are inaccurate, and the policy does not solve any of the tasks for the entirety of training.

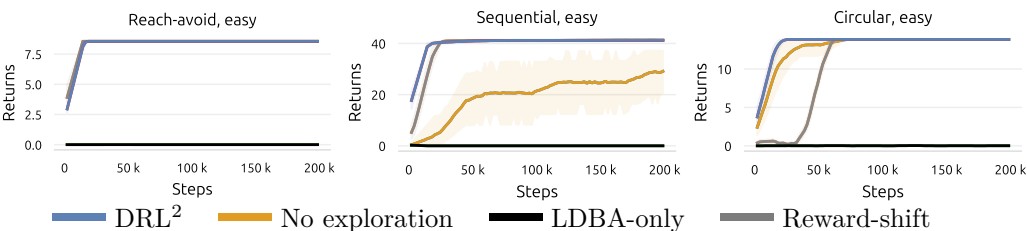

Figure 12: Evaluation of additional baselines in easy tabular tasks: *LDBA-only* trains a policy directly on the LDBA/MRP state, and *Reward-shift* shifts rewards to $\{-1/1\}$.

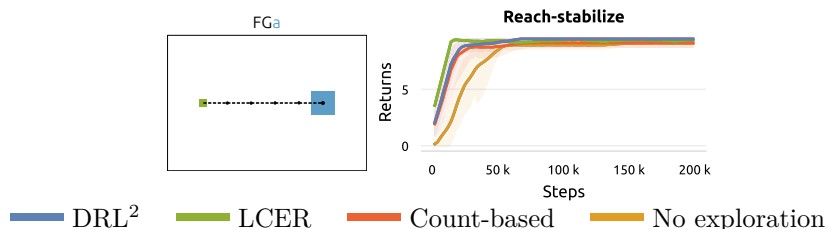

Figure 13: Evaluation of DRL$^2$ and baselines on the task $FGa$ in presence of jump transitions.

**Rewards** The eventual discounting framework manages to retrieve a proxy to an infinite-horizon objective by adopting a discount factor of exactly 1 for non-accepting transitions. As a consequence, the reward function for such transitions *needs* to be exactly zero. If this assumption is violated, the value function of a policy which does not visit an accepting transition infinitely often will diverge to (possibly negative) infinity. For this reason, from a formal perspective, the binary reward signal is crucial. However, in practical setting, one can apply an affine transformation to the reward signal, shifting it as $R'(b) = 2R(b) - 1 \in \{-1, 1\}$, thus punishing each timestep that does not transition to an accepting state. We evaluate this heuristic solution in tabular settings and observe interesting results in Figure 12. In simpler tasks (e.g., reach-avoid), this solution performs very well. This can be traced back to the fact that the Q-values are initialized to zero: while under $R$ they are not updated until an accepting state is visited by chance, in this case they will decrease as soon as the corresponding action is executed. This encourages the agent to prioritize yet unattempted actions, driving efficient exploration. This effect leads this baseline to match the performance of DRL$^2$ in simpler tasks, but is less effective in more complex tasks (e.g., circular), in which struggles to learn and is outperformed by the existing *No-exploration* baseline.

# I  Evaluation with $\epsilon$-transitions

This section evaluates application in presence of $\epsilon$-transitions. $\epsilon$-transitions occur in a subset of automata, in particular for stability specification; our method, as well as baselines considered, are compatible with theis existence. Thus, we now evaluate the task $FGa$ in reach-avoid tabular settings. This setup is similar to that of the corresponding experiment in recent work (Voloshin et al., 2023). Overall, as the structure of the LDBA is minimal, we observe that all considered methods perform reasonably well in Figure 13. In particular, DRL$^2$ and the count-based baseline slightly improve over the No-exploration baseline. LCER (Voloshin et al., 2023) works particularly well, as the task does not involve several substeps, but presents irrecoverable failures when "jumping" at the wrong time. We refer to Appendix E for a further comparison between DRL$^2$ and LCER.

# J  Evaluation under Stochasticity

**Stochastic initial state** Experiments in Figure 4 initialize the agent in a fixed state. Our evaluation does not however depend on this assumption. In order to verify this, we repeat the evaluation for the reach-avoid task, and sample the initial state uniformly from a rectangle in the leftmost part of the "corridor".

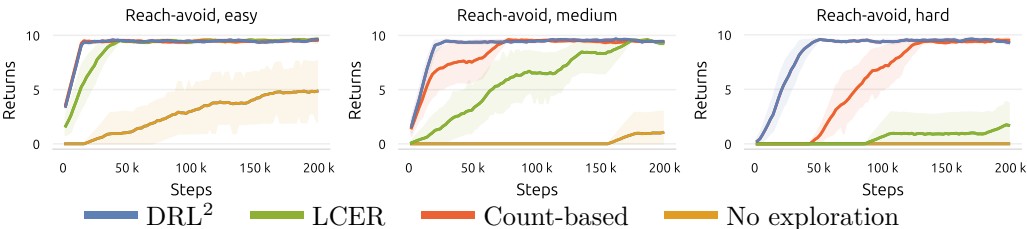

Figure 14: Evaluation of DRL$^2$ and baselines on the task $Fa\&G\neg b$ with stochastic initial states.

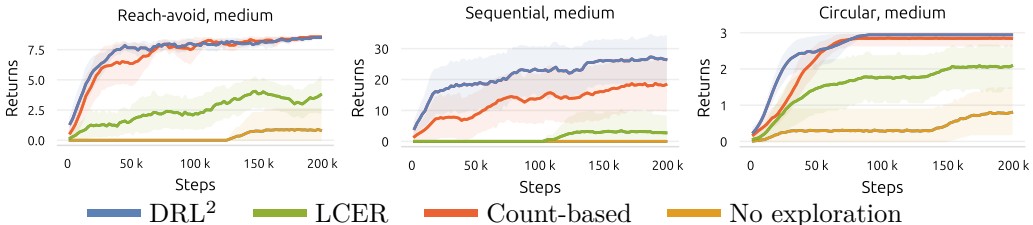

Figure 15: Evaluation of DRL$^2$ and baselines on medium tabular tasks from Figure 4 with sticky actions.

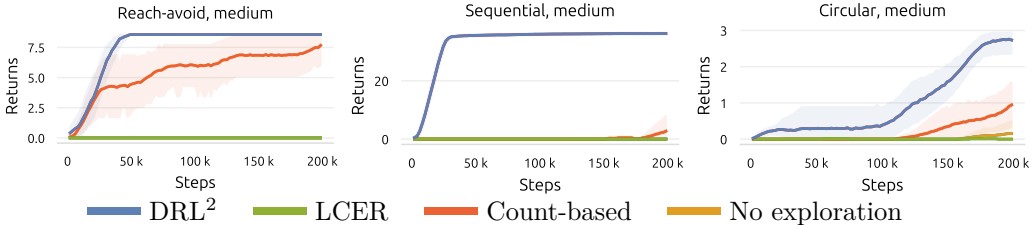

Figure 16: Repetition of tabular experiments while replacing Q-learning with SARSA.

In Figure 14, results are consistent with those in the original setting, with all methods being slightly more sample-efficient as they can be initialized in a more favorable position in some episodes.

**Sticky actions** We additionally repeat the tabular evaluation while injecting a difference source of stochasticity: *sticky actions*. Inspired by evaluation protocols on ATARI, with probability $p = 0.1$ at each step, we force the environment to ignore the agent's action and simply repeat the action from the previous time-step. This induces both stochasticity, and a slight partial observability, as the transitions now depend on previous actions as well. We report results in Figure 15. We observe that the overall trends from the original, deterministic variants are are largely reproduced, with an expected overall slight drop in performance. This suggests that the evaluated methods are at least in part robust to stochasticity.

## K  Evaluation under On-policy Learning

DRL$^2$ is in principle agnostic to the underlying RL algorithm, as it relies simply on reward shaping. We further validate this property empirically, by re-evaluating tabular problems of medium difficulty from Figure 4, and replacing Q-learning for SARSA Sutton & Barto (2018). In Figure 16, we observe that the overall sample efficiency decreases for all method, which is consistent with the on-policy nature of SARSA. Nevertheless, the relative performance of all methods remains consistent with the results obtained with Q-learning.

## L  Additional Metrics

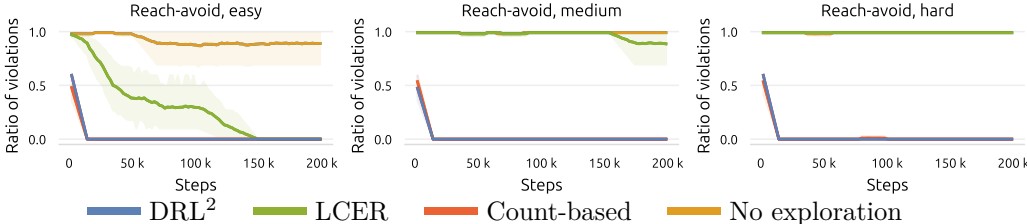

Figure 17: Evaluation of DRL$^2$ and baselines on the task $Fa\&G\neg b$ in terms of violations of the avoidance constraind $G\neg b$.

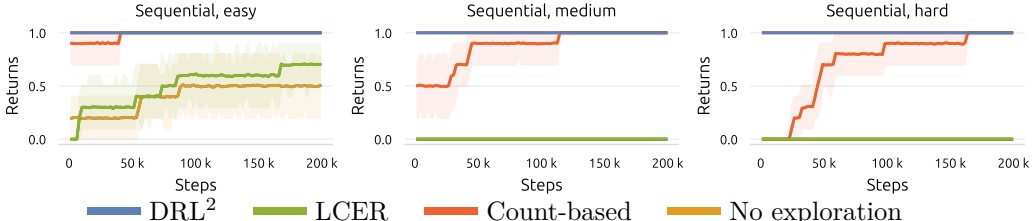

Figure 18: Evaluation of the likelihood of formula satisfaction in finite time for sequential tasks from Figure 4. DRL$^2$ satisfies the formulas from the earliest episodes, while its eventually discounted return grows during the initial stage of training. Intuitively, this means that the agent reaches an accepting state earlier and earlier.

**Contraint violations** While the standard evaluation in Figure 4 reports returns under eventual discounting, this section additionally reports the ratio of *training* episode that violate the avoidance constraints in the task $Fa\&G\neg b$. In Figure 17 we observe that, as expected, this metric is largely negatively correlated with evaluation returns: in order to maximize the likelihood of formula satisfaction, constraint violations need to be minimized. We additionally note one interesting trend: the count-based baseline quickly compensates the high rate of violations for a randomly initialized policy and quickly minimizes the violations. However, it requires more training steps to also reliably reach the specified zone.

**Formula satisfaction in finite time** Our empirical evaluation mainly relies on eventually discounted returns as a proxy for formula satisfaction. However, on some of the evaluation tasks, we can evaluate an additional metric. Namely, on sequential tasks without avoidance constraints, it is sufficient to visit an acceptance state once to satisfy the formula. While estimating the likelihood of visitation over infinite horizons remain intractable, we can evaluate the likelihood of a policy to visit an accepting state (and therefore satify the formula) within the episode length. We report an empirical estimate of this quantity in Figure 18, obtained by computing bootstrap confidence intervals of a satisfaction indicator across seeds. We observe that variance over seeds is limited: therefore, the plot closely resembles a *clipped* version of the metrics reported in Figure 4.

# M    Implementation Details

This section presents a thorough description of methods, tasks and hyperparameters. For the sake of reproducibility, we provide our full codebase [6].

## M.1    Environments and Tasks

Throughout the paper, experiments involve simulated environments.

### M.1.1    Tabular Environments

The environments considered in tabular settings (Figure 4) are variants of the common gridworld, that is a 2D grid, with one cell being occupied by the agent. The action space is discrete and includes 5 actions, four of which allow movement in each of the cardinal directions, and a single no-op action. The observation space contains $x$ and $y$ coordinates of the agent and is therefore 2-dimensional. While dynamics are shared, the labeling function from MDP states to atomic propositions and task-dependent details differ across tasks and are reported below. We note that each task is also represented in the rightmost column of Figure 4, although not to scale for illustrative purposes.

**Reach-avoidance** Reach-avoidance is evaluated in a gridworld without obstacles (Figure 4, top). The agent is initialized at one end of a thin corridor (of unit width). The other end of the corridor is not bounded;

---

[6]github.com/marbaga/drl2

however, the AP $a$ evaluates to `true` in each cell of the corridor beyond a fixed distance from the starting cell. The AP $f$ evaluates to `true` anywhere outside the corridor. The formula evaluated is $Fa\&G\neg b$. The difficulty of the task can be scaled by increasing the distance to the zone to reach (7, 9 and 11 for easy, medium and hard task variants, respectively). Episodes have a duration equal to the minimum number of step to reach the final zone, plus 10 steps.

**Sequential**    The second task in Figure 4 also takes place in a gridworld without obstacles. Contiguous $7 \times 7$ squares are aligned horizontally. The agent starts at the center of the leftmost square; in each square to the right, a different AP evaluates to `true`, in alphabetical order. In the first square to the right of the starting one, $a$ evaluates to `true`, in the second $b$ evaluates to `true` and so on. Difficulty can be scaled by specifying longer formulas: $F(a\&XF(b\&XFc))$, $F(a\&XF(b\&XF(c\&XFd)))$ and $F(a\&XF(b\&XF(c\&XF(d\&XFe))))$ are used, respectively, for easy, medium and hard tasks. Episodes have a fixed length of 70 steps.

**Circular**    Circular tasks (Figure 4, bottom) present 5 contiguous $7 \times 7$ squares, aligned as a cross. The central zone represents an obstacle: it is labeled to the AP $e$, and the agent cannot reach its central $6 \times 6$ cells. The other 4 zones are labeled as $a, b, c, d$ in counterclockwise order, and the agent is initialized in proximity of the first one. The difficulty of the task can be controlled by scaling the number of zones involved in a loop: $GF(a\&XFb)\&\neg e$, $GF(a\&XF(b\&XFc))\&\neg e$ and $GF(a\&XF(b\&XF(c\&XFd)))\&\neg e$ for easy, medium and hard tasks, respectively. An episode lasts for 70 steps.

### M.1.2 Continuous Environments

Section 4 also provides an evaluation on continuous, high-dimensional environments. Since evaluation of LTL-guided reinforcement learning has traditionally been mostly restricted to simple tabular (Icarte et al., 2022) or low-dimensional (Hasanbeig et al., 2020; Voloshin et al., 2023) settings, we repurpose high-dimensional environments from safe and goal-conditioned reinforcement learning research.

**Fetch**    Experiments reported in the first row of Figure 5 were developed on top of the common Fetch robotic benchmark (de Lazcano et al., 2023), and are also available as part of our codebase. We evaluate two tasks, in which the agent controls the end effector position of a 7DoF robotic arm with 4-dimensional actions over episodes of 50 steps. `FetchAvoid` is reminiscent of the reach-avoidance task in tabular settings, and involves fully stretching out the arm while avoiding lateral movements (i.e., not entering red zones as shown in Figure 5). In this case, observations are 10-dimensional. In `FetchAlign` the agent has to interact with three cubes on one side of the table, and has to align them horizontally at the center of the table. In this case, observations include information on the cubes, and are 45-dimensional. The two tasks are specified, respectively, with the formulas $Fa\&G\neg e$ (reach the end of the table, and avoid lateral movements), and $F(a\&XF(b\&XFc))$ (position the first, second and then third block).

**Doggo**    Doggo is a 12DoF quadruped adapted from the most challenging tasks in SafetyGym (Ray et al., 2019). It is tasked with navigating a flat plane; observations are 66-dimensional, actions are 12-dimensional and the length of each episode is 500 steps. Similarly to other reach-avoidance tasks, `DoggoAvoid` involves navigation to a distant goal (shown in green in Figure 5), in a straight line, while completely avoiding detours. On the other hand, `DoggoLinear` involves navigating through a sequence of 2 circular zones. These two tasks are encoded through the following two specifications: $Fa\&G\neg e$ and $F(a\&XFb)$. Episode length is set to 500.

**HalfCheetah**    HalfCheetah (Towers et al., 2023) involves controlling a 6DoF robot in a vertical 2D plane and is a standard environments in the deep reinforcement learning literature. We define two tasks in this underlying environments. In `CheetahSequential` the agent is tasked with eventually reaching, in succession, 4 consecutive zones to its right, as encoded by the formula $F(a\&XF(b\&XF(c\&XFd)))$. Once the last zone is reached, the task is solved and the agent can stop. In `CheetahFrontflip` the agent is informed by the formula $GF(a\&XF(b\&XF(c\&XFd)))$, where each variable evaluates to `true` for a given range of angles of the main body of the robot, respectively when the Cheetah is in its default position, standing on its front legs, upside down, and standing on its back legs. As a result, the formula is satisfied by an infinite sequence of frontflips. Notably, this task can not be satisfied by finite trajectories.

## M.2 Methods and Algorithms

The core of our method computes an intrinsic reward and is therefore applicable on top of arbitrary reinforcement learning algorithms. Therefore, we will first discuss the implementation of the method itself, as well as relevant baselines, and then details involving reinforcement learning algorithms.

The proposed algorithm is detailed in Algorithm 1. Its computational cost is dominated by the closed form solution of the value in the Markov reward model (see Equation 7), which is cubic in the number of MRM states. However, since LDBAs for most common tasks in these settings are relatively small ($< 20$ states), the computational load is negligible, even when computing the expected value over the posterior via sampling.

The main hyperparameter introduced by $DRL^2$ is $\alpha$, which controls the strength of its prior as described in Section 3.2. It is set to $10^3$ in tabular settings, and to $10^5$ in continuous settings, where increased stability was found to be beneficial. The remaining hyperparameters for reward shaping are shared with the count-based baseline and are, respectively, the frequency of updates to the potential function (set to 2000 environment steps), and a scaling coefficient to the intrinsic reward, which was tuned individually for each method in a grid of $[0.1, 1.0, 10.0]$. We found a coefficient of 0.1 to be optimal across all tasks for both methods in tabular settings, while continuous settings benefit from a stronger signal and a coefficient of 10.0. Intuitively, higher

Table 1: Hyperparameters for Q-learning and Soft Actor Critic.

| Hyperparameter | Value |
|---|---|
| $\gamma$ | 0.99 |
| Buffer size | $4 \cdot 10^5$ |
| Batch size | 64 |
| Initial exploration | $2 \cdot 10^3$ steps |
| $\tau$ (Polyak) | $5 \cdot 10^{-3}$ for SAC |
| Learning rate | $3 \cdot 10^{-4}$ for SAC, |
| | 1 for Q-learning |

coefficients can lead to better value propagation during the exploration phase at the cost of stability in deep learning settings; however, we found their impact to be limited (especially in tabular settings), such that no task-specific tuning was required. The two remaining baselines (relying on LCER and no exploration techniques) introduce no additional hyperparameters on top of the learning algorithm.

The intrinsic reward is added to the extrinsic one and optimized under eventual discounting through one of two algorithms, depending on the setting. In practice, we rely on tabular Q-learning (Watkins & Dayan, 1992) with an $\epsilon$-greedy policy ($\epsilon$=0.1) and Soft Actor Critic with automatic entropy tuning (Haarnoja et al., 2018b). In the tabular case, strict adoption of eventual discounting (Voloshin et al., 2023) results in a largely uniform policy, as in most states no action would reduce the likelihood of task satisfaction. As a consequence, a prohibitive number of steps would be required during evaluation for computing cumulative returns. For this reason, our practical implementation of tabular Q-learning *does not* apply eventual discounting, although this would be possible given a much larger computational budget. Our SAC implementation is partially based on that provided by Huang et al. (2022). Hyperparameters for each algorithm were tuned to perform best when no exploration bonus is being used, and are reported in Tables 1. They are kept fixed across tasks.

## M.3 Metrics

Unless specified, the metric used to measure the methods' performance is cumulative return under eventual discounting, that is $R = \sum_{i=0}^{N} \gamma^i$, where $\gamma = 0.99$ and $N$ is the number of visits to an accepting LDBA state within an episode. In the limit of $N \to \infty$, this metric is a proxy for the likelihood of task satisfaction (Voloshin et al., 2023). All curves report mean performances estimated across 10 seeds, and shaded areas represent 95% simple bootstrap confidence intervals. They undergo average smoothing with a kernel size of 10.

## M.4 Tools

Our codebase[7] mostly relies on `numpy` (Harris et al., 2020) for numeric computation and `torch` (Paszke et al., 2017) for its autograd functionality. Furthermore, we partially automate the synthesis of LDBAs from LTL formulas through `rabinizer` (Křetínský et al., 2018).

---

[7]Available at github.com/marbaga/drl2.

### M.5 Computational Costs

In principle, the distillation of the intrinsic reward according to $DRL^2$ depends on the size of the LDBA size. As we perform value estimation in the MRP built from the LDBA by computing the closed form solution of the Bellman equation, the complexity of this operation is cubic in the LDBA size. This remains tractable for reasonably sized LDBAs with $< 100$ states, and the computational cost is in practice negligible for the LDBAs in our empirical evaluation. For very large LDBAs, sampling-based alternatives for approximate value estimations should be considered instead.

When considering the training times of the underlying RL agent, all methods have comparable runtimes within their setting (tabular or continuous). Each experimental run required 8 cores of a modern CPU (Intel i7 12th Gen CPU or equivalent). Each run in tabular and continuous settings required on average $\sim 30$ minutes and $\sim 150$ minutes, respectively.

