# OpenReview forum: "Directed Exploration in Reinforcement Learning from Linear Temporal Logic"
_TMLR — Accepted by TMLR_

### Review · Reviewer_GUGh · 2025-01-12

**Summary Of Contributions:**

The paper introduces a novel reward-shaping approach based on LTL task specifications. The LTL specification is transformed in a Limit Deterministic Büchi Automaton that is combined with the environment MDP to a product MDP. Based on observed trajectories a value function can be computed that can be used to compute an "intrinsic reward" which is added to the sparse rewards from the product MDP.  The reward-shaping approach is evaluated in discrete and continuous spaces and compared to relevant baselines. The reward shaping leads to faster convergence and partially to enabling task solving.

The paper is well written. However, the abbreviations (e.g., RL, LTL, GridWorld) are not consistently used and some figures are not consistent with the template by not spanning the full column.

**Audience:**

Yes

**Claims And Evidence:**

Yes

**Requested Changes:**

- The more common term for similar methods would be goal-conditioned instead of directed reinforcement learning. Directed can be viewed as strong guidance as present in action masking methods. Please engage with this literature [1,2] and the goal-conditioned RL literature [3-5] more and clarify the distinctions in the related work section (and introduction).
- Related work:

a) As stated in the previous comments, some very related literature is not well enough discussed, which should be improved. Additionally, the related work section does not clarify the advantages or distinctions of your approach in contrast to existing work. Please add this information

b) It is unclear what the criteria are for the extended literature review and most references are presented in the main paper anyway. So please merge this in the main content.

- Discussion:

a) The limitations are not sufficiently discussed. Please answer the following questions: What interpretation limitations are present due to your experimental setup, i.e., what could be issues on other tasks? What if tasks are more stochastic than your current selection? How would more uncertainty, e.g., in the evaluation of atomic predicates, affect your approach? How does the potential mismatch of your prior distribution assumption with reality affect the approach? Where do you see the main weaknesses ofyour approach for future extensions or applications, e.g., what if parts of the task specification have to be always ensured (safety/avoid unsafe) and others are less relevant, how would you approach that, etc.?

b) The message of your last sentence is not clear. Please rephrase.

c) I would suggest you restructure the Discussion to a clear discussion section that mainly addresses limitations and gives ideas how they can be mitigated. After that, you can have a clear conclusion paragraph. Right now the information is a bit convoluted.

- Doesn't the virtual sink state change the task notion for cyclic tasks? How does this affect the resulting policies?
- "DRL2 drive deep.." -> You mainly show convergence rates evidence for that. Did you also check other metrics, e.g., how often irrelevant states are visited?
- the count-based and no exploration baseline could be introduced in a bit more detail or with more clarity (1-2 sentences) in the main part of the paper. Right now the description is vague
- the discrete environments seem to be overfitting for one specific initial state. Is this correct? Also, the sequential task seems to have complicated specifications that could be reduced to F(c). Why did you not place the colored fields somewhere in the grid?
- Can you comment on your hyperparameter tuning? I would highlight the information that \alpha is an additional hyperparameter for DRL2  in the main text and that you use the same hyperparameters for all approaches.
- The first sentence in the second paragraph of 4.2. is unclear and potentially incorrect since the continuous results seem to be significantly noisier.

Minor comments:
- The background section is very well written.
- You could make it a bit more obvious that R_{intr} is the reward shaping

References:

[1] Theile, Mirco, Lukas Dirnberger, Raphael Trumpp, Marco Caccamo, and Alberto L. Sangiovanni-Vincentelli. "Action Mapping for Reinforcement Learning in Continuous Environments with Constraints." arXiv preprint arXiv:2412.04327 (2024).

[2] Roland Stolz, Hanna Krasowski, Jakob Thumm, Michael Eichelbeck, Philipp Gassert, and Matthias Althoff. Excluding the Irrelevant: Focusing Reinforcement Learning through Continuous Action Masking. NeurIPS 2024

[3] Beyazit Yalcinkaya, Niklas Lauffer, Marcell Vazquez-Chanlatte, and Sanjit A. Seshia, Compositional Automata Embeddings for Goal-Conditioned Reinforcement Learning. NeurIPS, 2024

[4] David Lindner, Matteo Turchetta, Sebastian Tschiatschek, Kamil Ciosek, Andreas Krause. Information Directed Reward Learning for Reinforcement Learning.  Part of Advances in Neural Information Processing Systems 34 (NeurIPS 2021)

[5] Directed Exploration for Reinforcement Learning. Zhaohan Daniel Guo, Emma Brunskill. https://arxiv.org/abs/1906.07805

**Strengths And Weaknesses:**

Strengths:

- The reward shaping method proposed is relevant and seems to be very performant
- The evaluation is overall through and encloses many different settings

Weaknesses:

- The discussion, especially the limitations, is insufficient, i.e., too short, very limited information
- The literature review should be improved, i.e., the distinction or novelty in contrast to other approaches is not clear, especially to goal-directed RL

---

> ### Author Response · Authors · 2025-03-03
>
> We would like to thank the reviewer for the thorough feedback they provided. We are glad to hear that the reviewer finds several sections to be well-written and the evaluation to be thorough, and we have integrated all suggestions to improve the manuscript as a whole. We refer to the points below, and to the updated manuscript (changes in red).
>
> **Discussion**
>
> We have overhauled and extended the discussion (Section 6) as suggested: we reserved a paragraph for limitations, followed by a conclusion. We have also rephrased the last sentence to be more concise. On the topic of limitations, we have significantly extended the discussion. We provide a concise answer to the questions here, and refer to the revision for a broader and more detailed discussion.
> - We believe that our method is not suitable for certain families of task, including very long-horizon tasks and those not requiring significant exploration of the underlying MDP.
> - Uncertain evaluation of predicates is also a potential issue, as the bulk of work in LTL-guided RL assumes a ground truth labeling function. Our work does the same, and does not directly tackle inference of this labeling function.
> - Empirically, we did not find mismatches between prior and evidence to be problematic: in fact, the prior is generally poorly representative of the occupancy of the initial policy. Nonetheless, the posterior update strategy is in principle able to reduce this gap. We believe that the algorithm could however be slow to converge in adversarial cases.
> - We believe that the greater limitations are those outlined above, and mostly arise in specific classes of tasks, and in the absence of an accurate labeling function.
> We remain available to discuss these points, and answer any further question.
>
> **Related works**
>
> Thank you for suggesting further literature. We have integrated the related works in Section 5 with those in the Appendix, and the further references you have provided. Furthermore, for each relevant research direction, we now explicitly discuss the distinctions and advantages of our approach.
>
> **“Directed” exploration**
>
> Although “directed” is an overloaded term, we believe that it is a good descriptor for our algorithm, as the reward terms we distill “directs” the agent towards an accepting state in the LDBA. “Goal-conditioned” is a closely related term, but it is often coupled with multi-goal setups, while our agent specializes to each logic formula. We have added a paragraph to related works to discuss this differences and further elaborate on naming.
>
> **Virtual sink**
>
> For tasks already involving an avoidance constraint (e.g., the circular task in Fig. 4), the virtual sink does not affect the algorithm in any way, as the LDBA already features a sink state. For other tasks, the addition of a virtual sink does not affect the task definition, the product MDP, not the evaluation. It only alters the intrinsic reward distilled by our method (and hence, the learned policies). Intuitively, if a virtual sink is imagined, the value estimation procedure (Eq. 7)  will consider a non-zero chance of transitioning to the sink, and preventing visits to the accepting state. This encourages induces non-zero intrinsic rewards (Eq. 5) in these tasks, which are important for better exploration, as we demonstrate in Figure 6.
>
> **Visits to irrelevant states**
>
> We can additionally log visits to “bad” states (e.g. violations of the avoidance constraint). This metric is largely correlated with returns, as each visit to states to be avoided induces a return of $0$ for the episode. We have added an illustrative plot in Appendix J.
>
> **Discrete environments and initial states**
>
> The tasks considered in the discrete case have a deterministic initial state, as the reviewer correctly points out. However, our method does not require this assumption. To show this, we observe consistent results when sampling the initial state uniformly from a set, as we report in the new Appendix J.
> To answer the second part of this comment, we believe that sequential tasks cannot directly be reduced to F(c). Each policy satisfying the current specification F(a&XF(b&XFc)) also satisfies F(c), but the inverse is not true. The specification requires the policy to specifically traverse all colored areas: it is not allowed to skip to the last one directly. The visual rendition includes 3 colored fields, one per atomic propositions. We would ask the reviewer to comment on whether this clarification answers their question.

---

> > ### Author Response · Authors · 2025-03-03
> >
> > **Other changes**
> >
> > - We updated figure placement and the use of abbreviations to conform to the style suggestions.
> > - We ensured that “reward shaping” is mentioned directly above the introduction of $R_\text{int}$ (Eq. 5).
> > - The baseline description in Section 4 was extended to describe them more precisely.
> > - We added a comment on hyperparameter tuning in the main paper, right below baselines.
> > - The first sentence in the second paragraph of 4.2 was rephrased as per the reviewer’s suggestion.
> >
> > We finally thank the reviewed for the actionable suggestions, which we have directly incorporated in the revision. We are happy to engage in further discussion, or clarify any point of our response.

---

> ### Comment · Reviewer_GUGh · 2025-03-12
> **Reply to author response**
>
> Thanks a lot for your answers and reviewed paper. In particular, the limitation paragraph covers my questions very well.
>
>
> I would like to clarify my previous comment "Also, the sequential task seems to have complicated specifications that could be reduced to F(c). Why did you not place the colored fields somewhere in the grid?":
>
> I'm referring to the rightmost column and the middle plot in Fig. 4. Here, the optimal policy of F(c) = the optimal policy of F(a&XF(b&XFc)) since to optimally get to c one visits a first and then b and then c. So, I was wondering why you did not distribute the a and b areas in a way that the optimal policy of your specification does not also solve F(c). Or am I not understanding your environment setup correctly? Are a, b, and c randomly set and the environment in Fig.4 is just an example?
>
> Thanks for explaining the meaning of "directed". It might improve the readability further if you state your notion of "directing" also when you introduce the name of your method and not only in the related work.

---

> > ### Author Response · Authors · 2025-03-13
> > **Reply to reply**
> >
> > Thanks for the additional clarification. You are absolutely right: in the sequential task from Figure 4, the optimal policies for the two specifications largely coincide. For $F(c)$, a policy going "around" the first few fields would also satisfy the formula, but in practice this solution is not found.
> > Placing the colored field elsewhere is definitely possible: in fact, we also evaluate two alternative variations of the sequential task, in which policies satisfying $F(c)$ and $F(a$&$XF(b \dots)$ are more distinct, namely in Figure 8 and Figure 9, right.
> >
> > We are glad to hear the limitation paragraph and the related works were well received; we are happy to present the notion of "directing" from the introduction in the next revision.

---

### Review · Reviewer_V9PM · 2025-02-08

**Summary Of Contributions:**

The title of this work is slightly misleading. The work primarily concerns how
reinforcement learning techniques can be used to construct policies that fulfill
objectives defined using LTL in an unknown MDP. This is still an interesting
problem, though it is not really about exploration in reinforcement learning per
se (note that otherwise a different selection of baselines and benchmarks would
have been necessary). In any case, the work shows how a reward signal can be
generated for the LTL specification that, together with Bayesian updating in an
episodic setting, asymptotically corresponds to (bounds on) the probability of
the LTL objective being satisfied in the MDP. (There is an analysis of asymptotic
convergence, though it does not provide explicit bounds on the convergence rate.)
In a handful of simple benchmarks, the method is demonstrated to generally yield
much faster convergence than e.g. counterfactual experience replay from prior
work by Voloshin et al.

**Audience:**

Yes

**Broader Impact Concerns:**

I don't have any concerns here.

**Claims And Evidence:**

Yes

**Requested Changes:**

I'd like to see the approach to learning continuous environments clarified -- I
don't see a discussion here (maybe I missed it) so I am assuming that the
states of continuous environment are simply discretized as a grid. (Hence, in
part, my concern about scaling.) I would like to see this made explicit if so.

Otherwise, a handful of things could strengthen the work, though they are not
critical: for one, an environment that requires more exploration than the U, such
as a maze, would be an interesting demonstration of the ability of the method
to actually drive exploration. For two, bounds on the convergence rate and
time complexity (beyond merely asymptotic convergence) would be helpful.
I guess these would just be polynomial in the number of states, but it would be
good to show this if so.

**Strengths And Weaknesses:**

The method is relatively simple and seems to perform well. I am glad to see some
experiments in continuous domains: the main downside of the method is that the
representation of the transition kernel seems to be essentially tabular, using an
explicit distribution on transitions between pairs of states. I'll also note that
the environments are still pretty simple as far as exploration is concerned.
They were, nevertheless, sufficient to demonstrate improvement, so this is fine.

---

> ### Author Response · Authors · 2025-03-03
>
> We would like to thank the Reviewer for their feedback and the suggestions! We will address the comments one at a time.
>
> **Continuous environments and scaling**
>
> The transition kernel that our method estimates is in fact not the transition kernel of the underlying MDP (i.e., a continuous MDP involving low-level information such as joint positions of a robot). As the reviewer suggests, this would require some discretization scheme, and would represent a significant limitation.
> Instead, our method estimates the transition kernel over the LDBA that is used for tracking formula satisfaction. The LDBA is inherently discrete, as there is a finite number of atomic propositions. As such, we do not attempt to estimate transition probabilities over low-level states, but only over high-level LDBA states. For this reason, our method is easily applicable irrespectively of the details of the underlying MDP (e.g., for reach-avoid tasks, the formula is the same for both continuous and discrete states; hence, the LDBA is the same, and its transition kernel has the same size). Intuitively, DRL$^2$ guides exploration by rewarding transitions over the LDBA, not in the underlying MDP. In the reach-avoid shape, for instance, the agent may not be rewarded at each step, but will receive a negative reward as soon as it violates the avoidance constraint.
> We hope that this clarifies why our method remains applicable to continuous environments, without any fundamental restriction. We have added this clarification to Section 3.1.
>
> **More exploration**
>
> We appreciate the reviewer’s suggestion. Our evaluation avoids standard mazes as the most straightforward specification (“reach the exit”) does not have sufficient structure to distill a meaningful intrinsic reward. However, if the specification includes high-level hints on how to navigate the maze, DRL$^2$ could use this information. We therefore present a new evaluation on a sequential task, with a low-level MDP resembling a maze (Appendix D). We observe that results are consistent with those in the standard sequential task, as the only difference is in the underlying MDP dynamis and labeling function.
> We concur that harder exploration problems would be an interesting benchmark, and this is the motivation between our evaluation in continuous settings, which we would consider more complex than a 2D U-maze. For instance, cheetah-flip would require the robot to perform a complete frontflip before observing ground-truth reward; similarly, fetch-align would require the manipulator to position all cubes in a straight line. In the first case the state space is 17-dimensional and the action space is 6-dimensional, while in the second case they are respectively 70- and 4-dimensional. In both cases, a policy would need to take the optimal action for $\approx$ 30 steps in a row to observe any ground truth reward.
> Additionally, we note that, as the agent acts in a product MDP, it is not sufficient to explore the state space of the underlying MDP, but the full state space of the product MDP must be considered.
>
> **Converge rates**
>
> We appreciate this suggestion, as we think that convergence rates would provide a more complete picture. Overall, we believe that our method would recover similar converges rate to those in standard RL under reward shaping, with additional terms depending e.g. on the diameter of components of the product MDP mapping to the same LDBA state. However, as analyzing convergence rates under reward shaping beyond logic-conditioned RL is in itself challenging, we would leave a thorough formal analysis for future work.
>
> We hope that this response addresses the reviewer’s comments, and remaining available for further discussion.

---

### Review · Reviewer_Gef2 · 2025-02-27

**Summary Of Contributions:**

This paper presents DRL2, a new method for directed exploration in Reinforcement Learning (RL) from Linear Temporal Logic (LTL) specifications. The core idea is to use the LTL formula’s Limit Deterministic Büchi Automaton (LDBA) as a Markov reward process (MRP). Then, by placing a Bayesian prior over LDBA transition probabilities, one can (i) produce a high-level value estimation over the automaton states, and (ii) convert these values into potential-based shaping rewards that guide exploration in the RL agent’s “product MDP.” Empirical results suggest that DRL2 can handle tasks that combine both sequential subgoals and avoidance constraints in more complex, high-dimensional domains than previous LTL-based RL works. The authors show that their approach yields significant improvements in exploration whenever the environment or task structure demands nontrivial trajectories to reach the automaton’s accepting states.

Recommendation: Accept with Minor Revisions

**Audience:**

Yes

**Claims And Evidence:**

Yes

**Requested Changes:**

- In the “Background” or “Product MDPs and Policy Optimization” section, introduce a Definition 2.5 that fully specifies the product MDP: M^×=(S^×,A^×,P^×,R^×,γ^×,μ_0^×).Make it explicit that 𝑅^×(𝑠,𝑏) is 1 if 𝑏∈𝐵^∗(accepting) and 0 otherwise (or clarify if it’s -1 outside acceptance).

- Consider or at least discuss the case 𝑅^×(𝑠 ,𝑏)=−1 outside acceptance to see whether it expedites exploration. A brief experiment or mention of potential side effects (like “rushing” to acceptance) would clarify the trade-offs.

- The paper states the MRP is a high-level approximation and might lose information about the environment. Add a small experiment or at least a more detailed discussion showing how ignoring environment states can lead to suboptimal policy updates in certain corner cases.

- Section 3.2 covers the intuition of a symmetric Dirichlet prior. A short example or pseudocode could help illustrate exactly how the posterior is updated each time data arrives and how the authors sample the MRP transitions for value computation. Although the text references the Dirichlet formula, a quick demonstration (e.g., small LDBA with 2–3 states) would improve accessibility.

- In the main exposition, the authors consistently use 𝑅^× and 𝛾^×  for the product MDP. However, a few paragraphs define the “eventual discounting scheme” with slightly different notational approaches (e.g., Γ_𝑖). To streamline reading, double-check that all definitions align clearly.

- In Section 3 (or Implementation Details), add a short statement on why certain reward scaling coefficients (like 0.1 in tabular tasks, 10.0 in continuous tasks) work well and how they influence training stability or final performance.

- The paper states that the approach can introduce “slight non-stationarity” in the reward shaping signal, since the Bayesian posterior is updating. It might help to quantify or at least discuss how severely this might affect off-policy vs. on-policy algorithms.

- In the "limitations" of section 6, authors might also want to mention that:
          - Dependence on correct LDBA: DRL2 can be sensitive if the LDBA construction is large or suboptimal (this was mentioned in the Outlook, but it’s also a limitation).
          - Potentially unrealistic prior: The Bayesian approach can be powerful, but might rely on a prior that could mislead exploration in some tasks if not set carefully.
          - Fully or partially stochastic environments: The current experiments are mostly deterministic in the discrete case.

- The paper includes “No Exploration” (no shaping) vs. “Count-based” vs. “LCER” vs. “DRL2.” But they do not show a “purely MRP-based RL” scenario (where you ignore environment states altogether), nor a “−1 outside acceptance” scenario. The discussion in Appendix F + G shows different ways of building or initializing the kernel (empirical, partially informed, etc.). They do highlight that prior knowledge about the LDBA structure is crucial—DRL2 intentionally encodes knowledge that “the automaton might transition to any next state with uniform probability at the start.” That is prior knowledge. Indeed, a “good” exploration algorithm typically tries to minimize reliance on hand-crafted priors. DRL2 does rely on the knowledge of which LDBA states are reachable from which other states, plus the hyperparameter 𝛼. The authors do some ablations (varying 𝛼), but one could argue for further experiments with weaker or partially incorrect priors. Hence, it would be interesting if the paper had a dedicated section showing “No MRP,” “Only MRP-based shaping with no environment details,” or “Using −1/0 as a reward scheme.” But that is not included.

- The paper only compares DRL2 with LCER, a count-based baseline, and “no exploration” conditions. Other curiosity or optimism-driven approaches (e.g., intrinsic curiosity modules or random network distillation) might also be adapted to LTL. A short discussion on how DRL2 relates to these would be beneficial.

- In appendix A, if there is a notion of “time t” at which Γ_t≈0, it may be helpful to mention or suggest how large t typically is to yield the same policy in both \tilde M^×and M^×. Even a theoretical or empirical bound might be insightful for practitioners.


Typos:
- Section 3.2: we adopt an partially symmetric prior ---> an partially

**Strengths And Weaknesses:**

Strengths:

- Addressing exploration challenges in RL settings such as multi-stage tasks, safety constraints, and repeated goals, is both necessary and nontrivial.
- The approach reframes the LDBA as an MRP (with eventual discounting), which is then leveraged for potential-based shaping. While reward shaping in automata-based RL is not entirely new, the Bayesian perspective on transition probabilities—along with an informed prior—fills a gap in earlier logic-driven RL research. Shaping based on “high-level” value estimates from the LDBA can counteract reward sparsity and direct exploration more effectively than prior baseline approaches.
- The experiments are systematically split between tabular (discrete) and continuous (high-dimensional) settings, demonstrating that DRL2 is not restricted to small toy problems. Also, the tasks exhibit varied forms of LTL formulas (reach-avoid, repeated visitation (GF), multi-step sequences (U and X), and so on) which validate the generality of the proposed method. The paper’s chosen environments form a reasonable test bed: 1) They cover both discrete and continuous settings. 2) They systematically scale LTL complexity and environment difficulty. 3) They demonstrate that DRL2 can handle typical RL tasks (navigation, manipulation, locomotion) under LTL constraints, showing promise for broader adoption.

- The paper offers a clear exposition of LDBAs, the product MDP construction, and the eventual discounting scheme. The derivations for potential-based shaping under eventual discounting are nicely explained, including references to prior work on policy invariance.


Weaknesses:

- Task Difficulty vs. Generalization: While the paper shows environments of varying sizes and complexities (e.g., gridworlds of different corridor lengths, continuous control tasks like Fetch and HalfCheetah), all tasks are still somewhat hand-crafted to highlight specific properties (reach-avoid, multi-step sequences, etc.). It would be interesting to see if DRL2 generalizes well to less structured or more chaotic environments—for instance, tasks that do not neatly separate into corridor vs. obstacle, or that involve a variety of sub-tasks simultaneously. As it stands, the suite focuses on “classical” logic tasks, which are helpful as controlled benchmarks, but don’t necessarily confirm how DRL2 would perform in truly open-ended or partially observable domains.

-  LTL Formula Design: The chosen formulas (Fa&G¬b, GF(a&XFc), etc.) indeed represent classic temporal motifs (reachability, avoidance, repeated visits). However, they do not show certain other patterns (e.g., complex nesting of temporal operators, mixing multiple U operators, heavily nested G/F forms). While the “suites” let the authors systematically scale up the minimal number of steps or the number of subgoals, it might be that real-world LTL tasks have more subtle structures. A broader variety of formula types (or multiple deadlines, concurrency constraints, etc.) might further test DRL2’s flexibility.

- Scalability and State Space: Although the paper does push beyond small toy settings to robotics tasks, it remains unclear how DRL2 scales if the LDBA itself becomes large (e.g., a formula with dozens of states). The authors do mention that typical LTL tasks yield relatively small LDBAs (< 20 states), so it may be a non-issue in practice. But if the environment is also massive (very high-dimensional), it might still be challenging. The method is presumably suitable for bigger grids or longer horizon tasks, but the paper’s largest discrete domain has episodes of maybe 50–500 steps. So the “scalability under extremely long horizons i.e. steps>1000” remains only partially tested. Even though the approach might remain feasible, it is worth clarifying or at least discussing if any modifications are required for stable learning in extremely long tasks.

- Imperfect Prior: The experiments rely on a Bayesian prior over LDBA transitions (Dirichlet), which is a “symmetric, optimistic” assumption. It’s certainly a valid design, but in truly unknown or “messy” tasks, one might question if this prior is overly optimistic or if it could lead to misdirected exploration. The authors do present ablations on the prior’s hyperparameter  𝛼. However, for certain tasks, an incorrect or imperfect prior might hamper performance or cause needless exploration. Additional tasks that illustrate the “prior mismatch” scenario might have been instructive.

- One-Step vs. Multi-Step LDBA Transitions: The paper acknowledges that in a real environment, multiple steps in the MDP might be required to induce a single LDBA transition (like going from one subgoal zone to another). The environment suite is still largely structured so that each subgoal region is fairly discrete. A more continuous or “scattered” environment, with partial observability or less direct transitions, could challenge the assumption that states are easily reachable.

- Testing with a different algorithm might provide additional evidence of DRL2’s algorithm-agnostic nature.

- In the reach-avoid tasks, failing the avoidance criterion immediately transitions to a sink state, effectively giving the agent a strong negative outcome. This helps with exploration in that domain, but it is a domain-specific design. In more general tasks (especially outside of gridworlds), “failure” might not be so clearly signaled. The paper could discuss whether DRL2 might struggle in tasks where the environment does not forcibly terminate upon failure, or if no well-defined “sink” LDBA state is reachable (e.g., in purely liveness tasks without avoid constraints).

- Many of the discrete tasks revolve around corridor or “room” setups that funnel the agent in a straightforward path. This is typical of LTL-based RL papers but can be seen as somewhat contrived. One might wonder how DRL2 performs in a more branching or maze-like environment—would the agent still effectively find the correct path given more branching choices and no single corridor?

- Although the continuous tasks (Fetch, HalfCheetah, Doggo) have different shapes—manipulation vs. locomotion—they still focus on fairly similar formula types (reach-avoid, finite or repeated subgoals). The authors do not explore, for example, tasks with nested ‘until’ operators or other formula structures that might yield more complex or less direct exploration demands.

- The paper explicitly notes that the 2D gridworld environments are fully deterministic. While this makes analysis simpler (especially for LTL), it doesn’t test how DRL2 handles stochastic transitions. In real applications, environment dynamics may contain significant randomness, which can alter exploration requirements (e.g., some states might be visited unpredictably). An additional set of stochastic gridworld tasks (random wind, slippery actions, etc.) could validate DRL2’s robustness to stochastic MDPs.


- The authors used cumulative return under eventual discounting as their metric to measure the method performance. This metric is not uncommon in LTL-based RL research, however, researchers seeking a direct measure of “policy satisfaction probability” or “percentage of episodes meeting the entire LTL formula” may prefer a binary satisfaction rate or an explicit horizon-based measure.

- Summing 𝛾𝑖 for each visit to an accepting state can help rank policies better (more visits to acceptance = higher return), it may not be as transparent as “the fraction of time the formula is ultimately satisfied.” Certain LTL formulas (like repeated “GF” type goals) do require multiple visits to acceptance, so counting repeated visits makes sense. But for simpler tasks where you only need to reach acceptance once or you need acceptance in a final region, the “count repeated visits” approach may artificially inflate the score. Moreover, this metric does not reflect whether the agent got “close” to acceptance but did not quite make it—non-accepting runs effectively get 0 aside from discount accumulations. It is informative to use a more granular measure, such as distance to acceptance or partial subtask completion. A more direct measure might be: “What fraction of episodes actually satisfy the LTL formula?”, or “What is the agent’s estimated probability of infinite repeated satisfaction?” Those would be more straightforwardly connected to logic satisfaction. By comparison, “eventual discounting” plus repeated visits may require additional interpretation.

- Although the authors compare against a few strong baselines (LCER, count-based shaping), there exist other exploration techniques in RL that might be adapted to LTL settings—e.g., intrinsic curiosity modules, hash-based exploration, or optimistic/pessimistic MDP approaches. A concise discussion of how DRL2 complements or differs from these more general exploration heuristics would strengthen the paper.

- Some mention of potential state space blow-up is given, but clarifying the computational cost when LDBAs become large (e.g., dozens or hundreds of states) could be more detailed.

- Currently, no direct experiment in the paper shows “the difference between using the full product MDP vs. using only the MRP.” The authors mention it conceptually: once you collapse environment states (𝑠∈𝑆) into a single LDBA state, you lose the detail of which environment state led there. The MRP lumps all transitions from (s,b) purely into a single next-b′ distribution. A potential experimental design could look like: Compare an approach that tries to plan or explore purely in the MRP (which sees only the LDBA states) vs. the normal RL in the product MDP. Show that certain distinctions in the environment get “blurred” in the MRP, leading to suboptimal or uncertain actions in some states. The authors do not have such an experiment in the paper; they only note the theoretical possibility. It could be a good extension if the authors want to highlight the magnitude of that “loss of information.”

---

> ### Author Response · Authors · 2025-03-03
>
> We would like to thank the reviewer for the detailed comments: we sincerely appreciate the time you invested in this review. We will address each requested change first, and merge or append answers to the comments in the “Strenghts and Weaknesses” section.
>
> ## Requested changes
>
> **Product MDP definition**
>
> We are happy to introduced Definition 2.5 (Product MDP) in the section on Product MDPs and Policy Optimization. We have changed notation to explicitly state that $R^\times=0$ for non-accepting states.
>
> **Negative rewards outside acceptance**
>
> Applying an affine transformation to the reward signal in order to provide a negative reward outside of acceptance is an interesting suggestion. We currently do not consider this option, as the discount factor is 1 outside of acceptance, and this transformation would allow the value function of an arbitrary policy to diverge to $-\infty$. In order to avoid this case, we could adopt a constant discounting, which however invalidates the bounds presented in [1], and thus complicate the connection between value function and likelihood of formula satisfaction. As the reviewer suggests, this solution would produce “rushing” or “myopic” policies. Finally, the value function of an agent optimizing a 1/-1 reward would still remain flat until an accepting state is stumbled upon by chance. We report this discussion, and validate this hypothesis experimentally in Appendix H. We find the suggested baselines to perform better than the “No-exploration” baseline in some tasks, in part due to a synergy with tabular Q-learning, but worse than DRL$^2$.
>
> **MRP-only baseline**
>
> We have also implemented this suggested ablation in Appendix H. We can confirm that an agent observing the MRP state only, instead of the full state of the product MDP, fails to satisfy even simple specifications, as it lacks access to necessary information for decision making.
>
> **Illustrating posterior update**
>
> As suggested, we have added the explicit posterior update formula between Equations 6 and 7. The value computation in Equation 7 is performed in closed form, as the MRP is known in full: thus, it does not require sampling transitions. We hope this clarifies this passage, and are happy to provide additional discussion if this is not sufficient.
>
> **Notation: $\Gamma_t$ and $\gamma^\times_t$**
>
> We have reviewed the use of $\Gamma_t$ and $\gamma^\times_t$, which we would like to clarify. $\gamma^\times_t$ represents the per-step discount factor, while $\Gamma_t$ is the product of per-step discount factors encountered so far along a trajectory. We however also referred to $\Gamma_t$ as $\Gamma^\times_t$ in one instance, which we now corrected. Thank you for finding this inconsistency!
>
> **Scaling coefficients**
>
> We have added the following short statement on reward scaling coefficients in the Implementation Details:
> *Intuitively, higher coefficients can lead to better value propagation during the exploration phase at the cost of stability in deep learning settings; however, we found their impact to be limited (especially in tabular settings), such that no task-specific tuning was required.*
>
> **Introduced non-stationarity**
>
> We have added the following paragraph discussing the impact of non-stationarity in a reworked Section 6:
> *[...] as several other exploration schemes, we note that DRL$^2$ introduces a slight non-stationarity in the reward signal, which needs to be addressed by the underlying learning algorithm and might be problematic in very stochastic environments, especially for off-policy algorithms. Nevertheless, this is a common issue of intrinsic exploration methods, which is in practice often neglectable.*

---

> > ### Author Response · Authors · 2025-03-03
> >
> > **Limitations: incorrect LDBAs, unrealistic priors, stochastic environments**
> >
> > As the reviewer points out, our evaluation largely aligns with standard benchmarks in logic-conditioned RL. As such, it does not directly involve highly stochastic and partially-observable environments, in which the LDBA might be estimated incorrectly, or the prior could be completely unrealistic. We believe that scaling this evaluation to these domains would be interesting, but these experiments would be fundamentally limited by the underlying algorithms (namely the standard RL algorithm and the eventual discounting scheme). Reinforcement learning algorithm can be brittle in highly stochastic environments, and the reward/discounting scheme we adopt strongly relies on a known labeling function, which would not be accurate in partially observable domains. As soon as the problem of inference of the LDBA is solved, however, we expect DRL$^2$ to be functional, as the reward signal it distills does not directly depend on the dynamics of the MDP. A suggested, we have added an explicit *Limitations* paragraph to Section 6, in which we mention issues such as suboptimal LDBAs, unrealistic priors and stochasticity, among others.
> > Nonetheless, we would like to refer to both existing and newly introduced empirical evaluations of these limitations, in particular with respect to (1) stochastic environments and (2) unrealistic priors. During this rebuttal, we have added an evaluation over stochastic variants of tabular environments. This is achieved, in practice, by adopting a stochastic initial state distribution, or “sticky” actions (a non-zero likelihood of overriding the agent’s action with the one executed at the previous time step). We refer to Appendix J for an evaluation in these settings.
> > Finally, we would also like to mention that Appendix F includes an evaluation under a strongly unrealistic prior. While Appendix G ablates scales the “strength” of the prior alone, Appendix F valuates a prior that instead fully ignores the LDBA structure, and is thus overly optimistic. We observe that it remains informative for certain tasks, it harms initial performance in others, but can eventually be corrected through posterior updates.
> >
> > **Curiosity, optimism and other exploration frameworks**
> >
> > We added a discussion hash-based exploration and optimism on page 8, footnote 4, to which we refer. In general, the application of exploration methods from RL to the decision process constructed from the LDBA is limited, and several other baselines can be designed. We choose a count-based instantiation as hashing is not needed in the discrete MRP state space, and optimism is not straightforward as the reward signal of the MRP is known and well-defined.
> >
> > **Time $t$ at which $\Gamma_t \approx 0$**
> >
> > We have added a footnote to Appendix A, which we refer to. In short, we clearly state that $\Gamma_t$ decays exponentially in the number of visits to the accepting states.
> >
> > ## Other comments
> >
> > **LTL formula design (discrete and continuous environments)**
> >
> > Our evaluation largely involves the formulas that have been considered in related works [1]. As the reviewer points out, these formulas span most LTL predicates, but do not densely cover the language. Nevertheless, as DRL$^2$ builds upon principled approaches that are compatible with LTL semantics in their entirety, DRL$^2$ remains appreciable to generic formulas, although its effectiveness might depend on the specifics. In general, we notice a trend suggesting that DRL$^2$ is comparatively more beneficial in more complex formulas, involving more atomic propositions and larger LDBAs (e.g. in the sequential task in Figure 4, which does not scale the structure of the formula, but still increases the number of atomic propositions).
> >
> > **Scalability over long horizons and large state spaces**
> >
> > The evaluation in this work is scaled up to the relatively small LDBAs generated from common specifications (up to $\approx 10$ LDBA states, and $\approx 10$ APs), and relatively long episodes (up to 1000 steps for continuous environments). As DRL$^2$ mostly extracts its intrinsic rewards from the LDBA, it would not constitute a bottleneck if the underlying MDP is very high-dimensional or requires long-horizon behavior. In this case, however, the core RL algorithm (e.g. SAC) would need to be compatible with long-horizon optimization, and remain stable with cumulative discount factors close to one.

---

> > > ### Author Response · Authors · 2025-03-03
> > >
> > > **One-step vs. multi-step LDBA transitions, “soft” failure**
> > >
> > > We agree that a “soft” labeling function and less direct transitions would be more realistic. As all methods we evaluate assume access to an accurate LDBA, and correct estimation of its transitions, their performance would degrade in these settings, and we expect a stronger impact as the environment gets noisier and less observable. A similar point stands for soft avoidance constraints: if the labeling function was, for instance, probabilistic, the base discounting scheme used for all methods we evaluate would need to be significantly changed. We do evaluate on tasks that do not involve avoidance constraints (e.g. Sequential tasks in Figures 4 and 5), but the underlying scheme for translation to an RL objective is not compatible with “soft” constrains for the time being.
> > >
> > > **Testing with a different algorithm**
> > >
> > > Thank you for suggesting this interesting ablation. Currently, our evaluation relies on Q-learning and SAC, suggesting the algorithm-agnostic nature of our method. In order to solidify this claim, we have added an evaluation in tabular settings with SARSA as an on-policy alternative. We refer to Appendix J for experimental results.
> > >
> > > **Corridor/rooms setup and “branching” environment**
> > >
> > > Introducing a “branching” option in the underlying tabular environment is definitely possible. For this reason, we adapt the Sequential task from Figure 4, and place the zones in the shape of a simple maze. As the LDBA is then informative of which transitions go in the direction of the ”exit”, we find that DRL$^2$ remains effective. We refer to Appendix D for experimental results.
> > >
> > > **Returns under eventual discounting and metrics**
> > >
> > > Unfortunately, as the reviewer acknowledges, likelihood of satisfying of general formulas is intractable in high-dimensional spaces, considering the infinite horizon setting. We appreciate the interesting suggestion to visualize the rate of formula satisfaction in a finite episode for simple tasks (e.g., those not involving global statements). We have thus added this evaluation for sequential tasks in Appendix L, which we refer to.
> > >
> > > **State-space blow up**
> > >
> > > We concur that DRL2’s complexity depends on the LDBA size. As we perform value estimation in the MRP built from the LDBA by computing the closed form solution of the Bellman equation, the complexity of this operation is cubic in the LDBA size. This remains tractable for reasonably sized LDBAs with $<100$ states. For very large LDBAs, sampling-based alternatives for approximate value estimations should be considered instead. We have added this discussion to Appendix K.5.
> > >
> > > **Typos**
> > >
> > > Thank you for finding the typo, it has been corrected!
> > >
> > > We hope we were able to address each one of the reviewer’s comments. If anything still needs to be addressed, we remain of course available for further discussion. Thank you for your feedback, which we believe was instrumental in improving the thoroughness and clarity of our work.
> > >
> > > **References:**
> > >
> > > [1] Voloshin et al., Eventual Discounting Temporal Logic Counterfactual Experience Replay, ICML 2023

---

> ### Comment · Reviewer_Gef2 · 2025-03-03
> **Thanks authors for enlightening responses**
>
> I have reviewed the comments and the new changes in the paper. I appreciate the authors' careful and thorough response to the comments. While I would like to continue the discussions, learn more about the authors' intuitions and observations, and suggest more clarifying experiments for the sake of my own curiosity, I believe this would be beyond the scope and claims of this work.
>
> The new experiments and added discussions either in the paper or the review responses are convincing enough for the current manuscript. Thus, I would like to say that I am happy and fully satisfied with the current state of the paper for publication.
>
> I want to thank the authors for producing solid science backed with systematic, clear evidence and sincere explanations of the scope and limitations. Logic-based interpretation to shape smarter reward function is indeed one promising direction in developing better RL algorithms, which also aligns with the biological brain.
>
> I enjoyed the paper. Keep up the good work, and good luck with your future research.

---

> > ### Author Response · Authors · 2025-03-13
> > **Reply to reply**
> >
> > We would like to thank the reviewer once again for their thorough feedback, and quick response to our updated submission.
> > We deeply appreciate the encouraging comments.

---

### Author Response · Authors · 2025-03-03
**General Response to Reviewers**

We would like to thank all reviewers for providing insightful comments and extensive feedback on our submissions.

We are glad to hear that our submission was found to be “well written” (GUGh and Gef22), to present a “relevant” (GUGh and Gef22) method that “performs well” (GUGh and V9PM), and is supported by an “overall” evaluation (GUGh, V9PM and Gef22). We also understand the suggestions to extend the discussions on limitations and related work, and to provide additional ablations and baselines.
We have thus updated our submission, including:
- several clarifications (Definition 3.5, an explicit posterior update formula on page 7, more baseline details on page 8);
- an extension of related works to include a discussion of directed explorations (Section 5);
- a refactoring of Section 6, with an extended limitation section;
- new experiments in a maze-like environment in Appendix D;
- new ablations and baselines in Appendix H;
- two additional evaluations in stochastic settings in Appendix J;
- an alternative evaluation relying on SARSA as a backbone RL algorithm in Appendix K;
- additional metrics, including violation rates and satisfaction over finite horizons (Appendix L).

All changes have been marked in red for ease of reviewing. We refer to individual responses for a detailed answer to each comment, and we are happy to answer any further comment.

---

### Decision · Action_Editor_zPxE · 2025-06-05

**Recommendation:** Accept as is

**Additional Comments:**

I apologize for the slowness in the final decision.  I was ready to make a decision without having all three reviewers make their final recommendation (as one was uncommunicative), but then they asked for more time right as I was going on vacation.

**Audience:**

Yes

**Audience Explanation:**

A principled approached to challenges in LTL-based RL has a significant audience.

**Claims And Evidence:**

Yes

**Claims Explanation:**

The reviewers generally felt that the original empirical demonstration was sufficient to support the claims of the paper.  They did note some minor concerns or questions about the generality of the approach, but the authors did a pretty comprehensive job in providing further details and empirical experiments addressing the reviewers questions and concerns.

They also well addressed its position in the broader literature with their revisions.